# Trans-toxin ion-sensitivity of charybdotoxin-blocked potassium-channels reveals unbinding transitional states

Hans Moldenhauer[1], Ignacio Díaz-Franulic[1], Horacio Poblete[2], David Naranjo[1]*

[1]Instituto de Neurociencia, Facultad de Ciencias, Universidad de Valparaíso, Valparaíso, Chile; [2]Núcleo Científico Multidisciplinario, Dirección de Investigación. Centro de Bioinformática y Simulación Molecular, Facultad de Ingeniería, and Millennium Nucleus of Ion Channels-Associated Diseases (MiNICAD), Universidad de Talca, Talca, Chile

**Abstract** In silico and in vitro studies have made progress in understanding protein–protein complex formation; however, the molecular mechanisms for their dissociation are unclear. Protein–protein complexes, lasting from microseconds to years, often involve induced-fit, challenging computational or kinetic analysis. Charybdotoxin (CTX), a peptide from the *Leiurus* scorpion venom, blocks voltage-gated $K^+$-channels in a unique example of binding/unbinding simplicity. CTX plugs the external mouth of $K^+$-channels pore, stopping $K^+$-ion conduction, without inducing conformational changes. Conflicting with a tight binding, we show that external permeant ions enhance CTX-dissociation, implying a path connecting the pore, in the toxin-bound channel, with the external solution. This sensitivity is explained if CTX *wobbles* between several bound conformations, producing transient events that restore the electrical and ionic trans-pore gradients. Wobbling may originate from a network of contacts in the interaction interface that are in dynamic stochastic equilibria. These partially-bound intermediates could lead to distinct, and potentially manipulable, dissociation pathways.
DOI: https://doi.org/10.7554/eLife.46170.001

*For correspondence:
david.naranjo@uv.cl

**Competing interests:** The authors declare that no competing interests exist.

## Introduction

Protein–protein interactions are essential for biological signaling. They are present in almost any aspect in the existence of a protein: from biogenesis to degradation. For most proteins, forming and disintegrating binding complexes is a significant part of their function. The binding energy determines the lifespan of a binding complex, extending from fractions of millisecond to the entire lifespan of the protein. The theory of absolute reaction rates proposes existence of an intermediate transition state in the reaction path (*Dill and Bromberg, 2011*). In this regard, there is a wealth of information on the events preceding the formation of the final protein–protein compound, which involves the formation of an encounter complex organized by a network of transient interactions funneling toward structural complementation and solvent displacement (*Frisch et al., 2001*; *Harel et al., 2007*; *Horn et al., 2009*; *Khabiri et al., 2011*; *Moritsugu et al., 2014*; *Paul et al., 2017*). Yet, the steps preceding the dissociation of the bound complex are little known. Because most physiologically relevant protein–protein interaction last longer than the millisecond time scale, detailed atomistic molecular dynamics (MD) simulation of dissociation events have been too challenging, computationally speaking. Moreover, transient intermediate states leading to dissociation must occur with very low probability, therefore, they would be poorly sampled (*Cavalli et al., 2015*).

Thus, most of our current understanding comes from MD simulations combined with enhanced sampling methods applied to simple ligands. These simulations show diverse transient states and pathways leading to the final dissociation event (*Pietrucci et al., 2009*; *Cavalli et al., 2015*; *Paul et al., 2017*; *Rydzewski et al., 2018*). Dissociation steps in more complex protein–protein interactions possibly follow the same trend but, perhaps more tortuous due to the larger number of contacts. Because induced fit and conformational selection certainly complicates even more the search of unbinding intermediaries (*Csermely et al., 2010*), a highly desirable condition to illustrate transition states in protein–protein interactions is to find a binding complex with simple stoichiometry and first order kinetics.

Here, we focused in the interaction between a scorpion peptide neurotoxin, charybdotoxin (CTX), and Shaker, a voltage gated potassium channel. The binding step of a single toxin molecule directly occludes the ion conduction pore, stopping $K^+$-ions conduction across the channel while toxin unbinding directly restores ion conduction (*MacKinnon and Miller, 1988*; *Miller, 1990*). Consistently with this mechanism, the crystal structure of the channel in the CTX-Kv1.2 complex is indistinguishable from that of this Kv-channel, or the toxin, alone, suggesting a key-lock binding type (*Banerjee et al., 2013*). Here we show in this mechanistically simple Kv-channel-CTX complex evidence for a pre-dissociation transitional binding conformation in which the pore acquires sensitivity to the trans-toxin environment. We propose that this trans-toxin sensitivity is due to time-unresolved events of partial toxin-unbinding belonging to the dissociation pathway.

## Scorpion toxin blocks the pore of potassium channels

Because the toxin binding occludes ion-conduction, the amplitude of Kv-channels mediated ionic current represents in high-fidelity the blocking and unblocking molecular events. As other cystine rich $\alpha$-KTX scorpion toxins, CTX binding is electrostatically aided and appears diffusion-limited, while their unbinding first order rate is enhanced by cell depolarization (*Anderson et al., 1988*; *Miller, 1990*; *Goldstein and Miller, 1993*). CTX plugs the pore, with a positively charged moiety acting as $K^+$ impersonator (*Figure 1A–D*) (*MacKinnon and Miller, 1988*; *Park and Miller, 1992*; *Goldstein and Miller, 1993*). This pore-occlusion mechanism was drawn because removal of *internal* permeant ions abolishes the voltage dependence of the dissociation rate (*Figure 1D*). It seems that potassium- and voltage-dependence are strongly linked. When CTX´s Lys27, a highly conserved residue among scorpion $\alpha$−KTX toxin family (*Figure 1A*), was replaced by non-charged residues, both, internal ion sensitivity and unbinding voltage dependence, disappeared. Because Lys27 was the only residues showing this property, its sidechain was identified as the potassium impersonator (*Park and Miller, 1992*) (*Figure 1A,D*). Later on, similar results were proposed for other cystine-knot peptide toxins, suggesting a remarkable example of mechanistic convergence with two different rigid scaffolds (*García et al., 1999*; *Naranjo, 2002*).

Searching for Lys27´s interacting partner in the channel pore, *Ranganathan et al. (1996)* observed that the external potassium dependence vanished when Tyr445 in the selectivity filter of the Shaker Kv-channel was mutated. Thus, permeant ions somehow mediated the functional coupling between these two proteins. The crystal structure of the CTX/Kv1.2 complex resolved in the MacKinnon lab, shows tightly bound and highly complementary interaction surfaces (*Figure 1A–C*; PDB:4JTA; *Banerjee et al., 2013*). A central structural feature is the proximity between CTX´s Lys27 amino headgroup and the carbonyl oxygen atoms in the vicinity of the Kv´s Tyr445, meanwhile the external $K^+$ binding site (S1) is empty (*Figure 1A–D*). Thus, the structure fulfills all the mechanistic aspects predicted before (*Swartz, 2013*).

Thus, the strong link between the voltage and potassium enhancement of the dissociation rate led to the idea that: a) Ions in the pore electrostatically repel Lys27, speeding the toxin dissociation (*Figure 1D*). b) In the toxin-blocked channel, membrane depolarization favor permeant ions moving into the pore, destabilizing the toxin binding (*MacKinnon and Miller, 1988*; *Park and Miller, 1992*). This explanation has counterintuitive aspects which leads us to the idea that, in fact, the voltage dependence of the dissociation rate may reveal the existence of transient unplugged states: While in the unblocked channel most of the transmembrane electric field should drop across the pore (sketched in *Figure 1E*), when the toxin is occluding the pore, the main resistance for the ionic current should be the toxin body itself, thus, only a small fraction of the electric field should drop across the pore (*Figure 1F*). According to this view, both phenomena, permeant ions stability within the pore and the toxin dissociation, should be voltage independent. Here, we examined the external

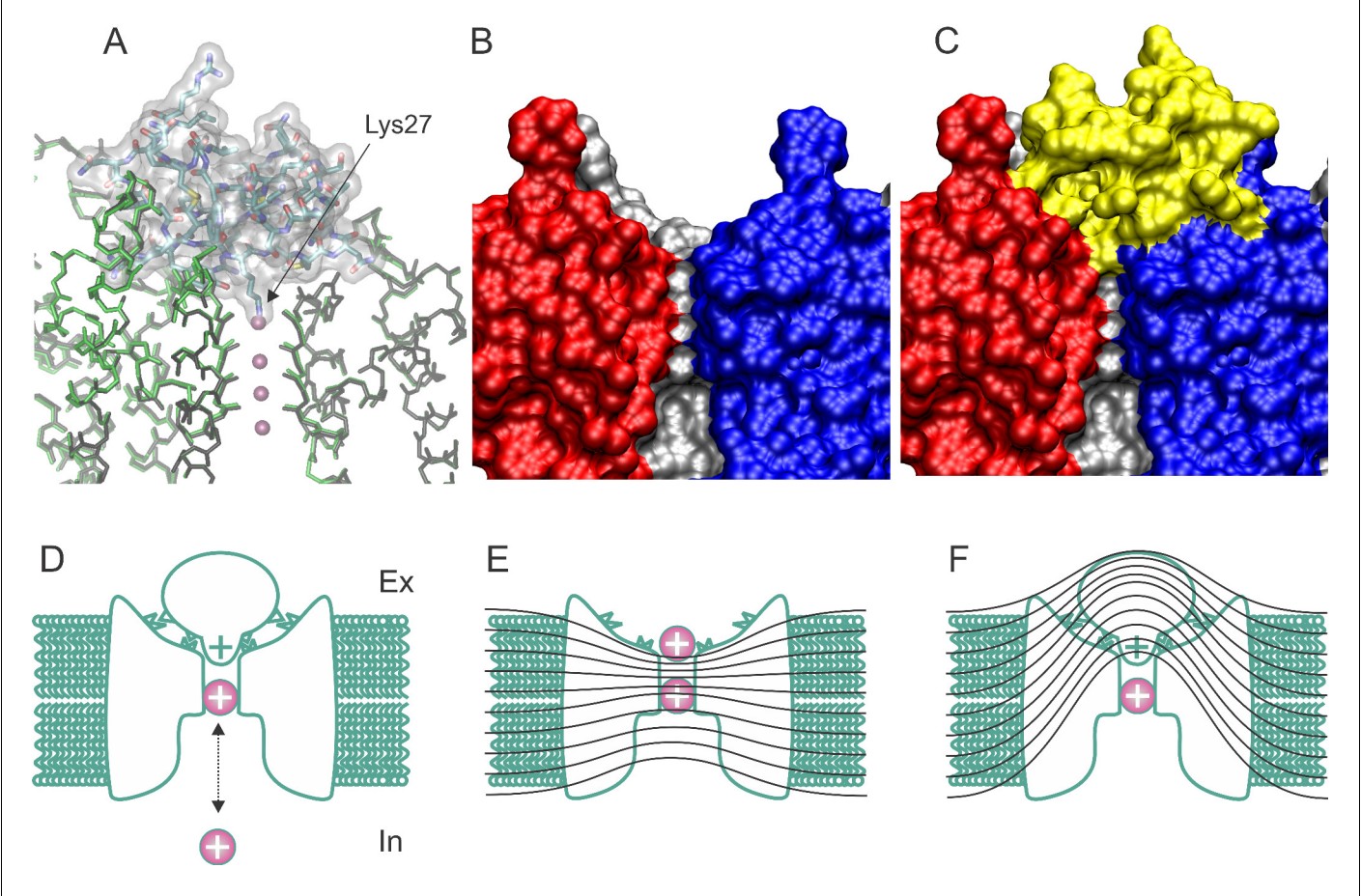

**Figure 1.** Structure and blockade mechanism of the complex CTX/Kv1.2–2.1 paddle chimera. (**A**) Superposition of the blocked and unblocked pore domain Kv-channel structures (PDB:4JTA and 2R9R, respectively). CTX is represented as a transparent surface with the Lys27 ε-amino group occupying the most external ion binding site (**S1**). Meanwhile both channel backbones are in stick representations (brown for 4JTA and green for 4R9R). Front and back subunits were omitted for clarity. The large superposition of both backbones indicates that the conformational effects of toxin binding are minimal. (**B–C**) Toxin and channel surfaces are complementary and tight. Surface representation of three pore domains of the blocked–Kv-channel (back, left and right) without CTX (**B**) and with CTX (yellow surface). Note that the external ion binding site is not empty in the toxin bound structure. (**D**) Cartoon representation of CTX plugging the pore at the external end of the Kv-channel (inspired from *MacKinnon and Miller, 1988*). The selectivity filter is in equilibrium with internal $K^+$ ions, and its occupation repels the bound toxin. (**E**) Schematic cartoon of the drop of the electric field along the pore of conductive channels and its alteration in the toxin blocked channel (**F**), such that some permeant ions movements do not cross the electric field. Lines represent iso-potential curves.

DOI: https://doi.org/10.7554/eLife.46170.002

potassium and voltage sensitivity of CTX binding to open and closed Shaker Kv-channels. CTX binding equilibrium is voltage sensitive in open channels only. The external potassium sensitivity as well as the voltage dependence, may be clues of the dynamic behavior of the toxin-channel interaction surface resulting from the unbinding/rebinding of separate local contact points anteceding the final unbinding step.

## Results and discussion

### Toxin-binding equilibrium is sensitive to the ion composition of the external solution

We first tested the effect of the external $K^+$ on CTX binding to the Shaker-F425G K-channel heterologous expressed in *Xenopus* oocytes. This Shaker variant exhibits high toxin affinity, with dissociation kinetics slow enough to allow for complete exchange of the recording solution, without significant toxin unbinding (*Goldstein and Miller, 1992*). *Figure 2* shows inhibition of $K^+$ currents by applications of CTX in normal ('High $Na^+$") and High $K^+$ recording solutions (See Materials and methods section). In high $Na^+$ solution, 0.25 nM CTX inhibited ~70% of the current, with an exponential onset of several seconds (*Figure 2A,C*). Consistent with a first order dissociation rate, upon toxin removal, the current recovers exponentially also (*Goldstein and Miller, 1992*). From the exponential fits to the blockade onset (*On*) and recovery (*Off*), we obtained the association ($k_{on}$) and dissociation ($k_{off}$) rate constants according to *Equations 1a and 1b*:

$$\tau_{off} = \frac{1}{k_{off}} \tag{1a}$$

$$\tau_{on} = \frac{1}{k_{on}[\text{Tx}] + k_{off}} \tag{1b}$$

where [Tx] is the toxin concentration. Thus, in high $Na^+$ $k_{on}$ and $k_{off}$ were 73 ± 9 $\mu M^{-1}s^{-1}$ and 0.0062 ± 0.0005 $s^{-1}$ respectively (n = 7) with the dissociation constant, $K_D$ = 0.086 ± 0.018 nM. Meanwhile, in High $K^+$ (*Figure 2B,D*), $k_{on}$ = 28 ± 11 $\mu M^{-1}s^{-1}$, $k_{off}$ = 0.013 ± 0.0014 $s^{-1}$, and $K_D$ = 0.55 ± 0.2 nM. This latter value represents a ~ 6.4 fold increment consistent with previous reports (*Goldstein and Miller, 1993*; *Ranganathan et al., 1996*). The reduction by ~60% in $k_{on}$ is expected if $K^+$ competes with CTX for the external mouth of the pore. However, because Lys27 is the only toxin residue mediating the $K^+$-dependent dissociation enhancement, the 2-fold increment of $k_{off}$ in 100 mM $K^+$ is unexpected (*Ranganathan et al., 1996*). Once the pore is plugged by the toxin, the selectivity filter should become uncommunicated from the external side, ergo, insensitive to the external $K^+$. In agreement with this, the structure of the channel-toxin complex shows a tight seal in the toxin-channel interaction surface (*Figure 1C*). Thus, the effect of $K^+$ on $k_{off}$ is unexpected.

To test for communication between the external solution and the selectivity filter in the toxin-blocked channel, we compared the $K_D$ and $k_{off}$ in the presence of external cations used typically to fingerprint $K^+$ channels selectivity filter (*Heginbotham and MacKinnon, 1993*; *Díaz-Franulic et al., 2015*). *Figure 2E* shows the ratios of $K_D$ for the test cations (with respect to that for $K^+$) for external solutions having $Rb^+$, $NH4^+$, $Cs^+$ and $Na^+$ as the main cation. For comparison, in log scale are the permeability ratios for Shaker obtained by replacing the test cation either externally (filled circles) or internally (*Heginbotham and MacKinnon, 1993*; *Díaz-Franulic et al., 2015*). Similarly, by comparing the relative $k_{off}$ enhancement we found that in external $Cs^+$ the rate constant is about identical to the observed in the High $Na^+$ control solution, meanwhile in $K^+$ and $Rb^+$, is about 2-fold and 3-fold, respectively (*Figure 2F*). Surprisingly, the ion dependent CTX-dissociation enhancement compares well with the one observed with *intracellular* ion applications to BK channels (*MacKinnon and Miller, 1988*). Such trans-bilayer effect was, and is still, considered landmark of ion-toxin interaction at the selectivity filter. Thus, the cation-permeation sequence agreement with $K_D$-order and the external ion-dependent dissociation enhancement, point to the existence of a form of communication between the CTX-dissociation enhancement site, in the selectivity filter, and the external solution in the toxin-occluded channel.

We looked in detail how CTX-binding and unbinding depend on the normal permeant cation, $K^+$, in a wide range of external concentrations (keeping ionic strength constant with $Na^+$). *Figure 3* illustrates how potassium affected $k_{on}$ and $k_{off}$ differently. While $k_{off}$ remains unchanged at $[K^+]<10$ mM, it raises abruptly at higher concentration. In addition, $k_{on}$ sharply decreases at $[K^+]>1$ mM, changing little at higher concentrations. For a quantitative evaluation of the effect of external $K^+$ ions on the enhancement site, we fitted the association and dissociation rates data of *Figure 3* to Langmuir isotherm that assumed that CTX binding and dissociation were antagonized and enhanced by $K^+$,

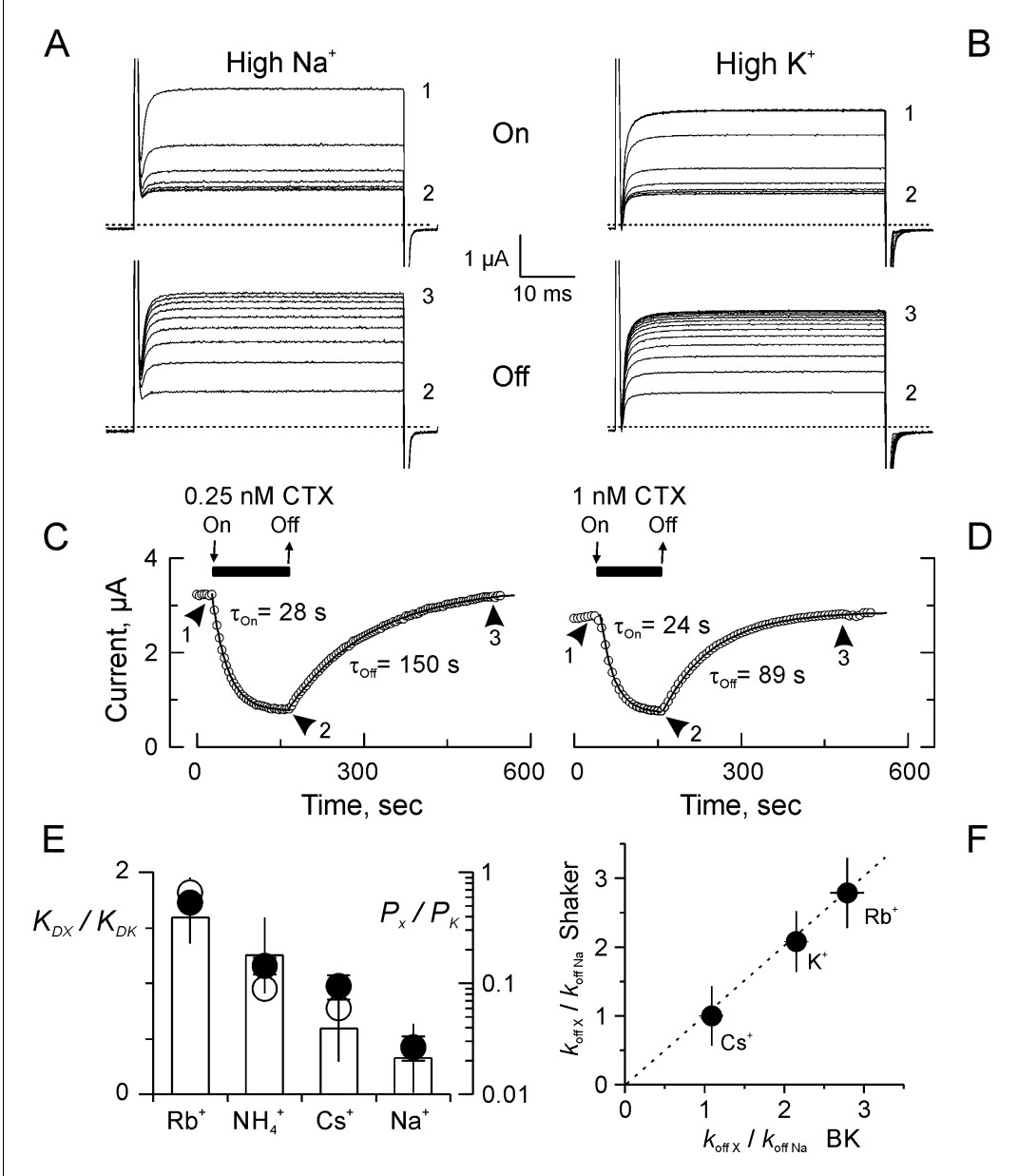

**Figure 2.** Kinetics and ion selectivity of CTX binding to Shaker-F425G channels in 'High Na$^+$' and 'High K$^+$'. (A–B) Application (*Upper panel*; '*On*') and removal (*Lower panel*; '*Off*') of CTX in High Na$^+$ (A) or High K$^+$ (B) recording solutions. To watch the progress of the perfusion, current traces elicited by 50 ms/50 mV voltage pulses applied every 3 s. Traces marked 1, 2 and 3 were taken: before toxin application, before removal, and at the end of the recovery, respectively. (C–D) Time course of the inhibition and recovery in High Na$^+$ (C) or High K$^+$ (D) recording solutions. CTX application is marked by the black bar on top of the time-course plot. Each point is the average current-amplitude in the last 10 ms of each trace. The 'On' and 'Off' phases of the experiments were fitted to a single exponential function, and the time constant for each fit is written near each time course. (E) Ion selectivity of the dissociation enhancement site. The $K_D$ relative to 'High K$^+$' (white bars) were obtained measuring inhibition when the main external ion was the test ion (See Materials and methods). Each value corresponds to the average (± SEM) of 3 to 7 oocytes. For comparison, the relative selectivity ratios for permeation in bi-ionic from are plotted in log scale, according to *Díaz-Franulic et al. (2015)*, filled circles, and to *Heginbotham and MacKinnon (1993)*, open circles. (F) Ion dependent dissociation enhancements, relative to control (*High Na$^+$*) external solution ($k_{off\,X}/k_{off\,Na}$), is near identical to the effect of internal cations in the dissociation of CTX from BK-channels reconstituted in lipid bilayers (*MacKinnon and Miller, 1988*). The discontinuous straight line has a slope of 1.
DOI: https://doi.org/10.7554/eLife.46170.003

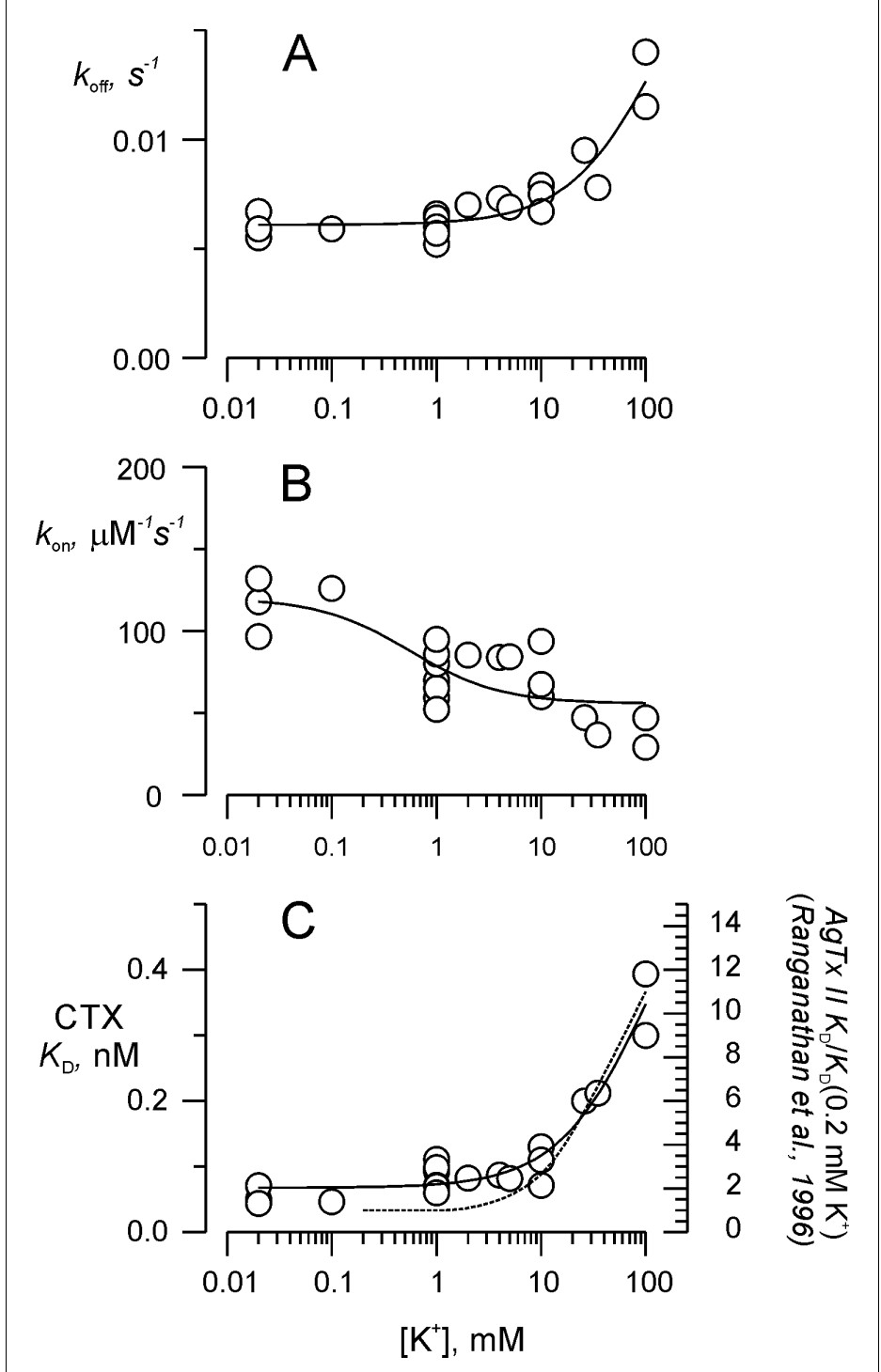

**Figure 3.** Potassium titration of the toxin dissociation enhancement site. (**A**) Potassium dependency of the dissociation rate, $k_{off}$, (**B**) association rate, $k_{on}$, and (**C**) dissociation constant, $K_D$. To compare with *Ranganathan et al. (1996)*, we show in (**C**) the fit of *Equation 2a* (solid line), in which $K_D$ replaces $k_{off}$, $K_{D0}$ replaces $k_{off0}$ and $K_{DK}$ replaces $k_{offK}$. Fit parameters were (± SE), $K_{D0} = 0.068 \pm 0.008$ nM, $K_{DK} = 0.65 \pm 0.18$ nM, with $K_{K1} = 108 \pm 60$ mM. For comparison, the dashed line on C represents AgTx II - $K_D$ as function of external $K^+$ from *Ranganathan et al. (1996)*. Values are individual experiments.
DOI: https://doi.org/10.7554/eLife.46170.004

respectively. The expressions used to describe how the apparent rate constants, $k_{off}$ and $k_{on}$, depend on [K$^+$] were the following:

$$k_{off} = k_{offK} + (k_{off0} - k_{offK})\frac{K_{K1}}{K_{K1} + [K^+]}\tag{2a}$$

$$k_{on} = k_{onK} + (k_{on0} - k_{onK})\frac{[K^+]}{K_{K2} + [K^+]}\tag{2b}$$

where $k_{off0}$ and $k_{offK}$ in *Equation 2a* are the dissociation rates in the absence and in the sole presence of external K$^+$, respectively. Similar meanings are for both association rates in *Equation 2b* $k_{on0}$ and $k_{onK}$. Meanwhile, $K_{K1}$ and $K_{K2}$ are the K$^+$ dissociation constants for the site that enhances CTX-dissociation and the one that antagonizes the toxin-association, respectively. Fitting *Equation 2a* to the data in *Figure 3A* gives $k_{off0}$ = 0.0061 ± 0.0002 s$^{-1}$, $k_{offK}$= 0.022 ± 0.007 s$^{-1}$ with $K_{K1}$ = 135 ± 107 mM. While the fit of *Equation 2b* to data in *Figure 3B*, gives $k_{on0}$= 120 ± 11 µM$^{-1}$ s$^{-1}$, $k_{onK}$= 56 ± 8 µM$^{-1}$ s$^{-1}$, with $K_{K2}$ = 0.56 ± 0.39 mM. Thus, these fits predict a ~ 9 fold increase in the dissociation constant after external K$^+$ replacement with Na$^+$. *Figure 3C* shows that this expectation holds for $K_D = k_{off}/k_{on}$ as function of external K$^+$. For comparison, the discontinuous line shows the AgTx II-K$^+$ dependence from *Ranganathan et al. (1996)*. This is not a purely competitive scheme in which only the toxin association is sensitive to the external competitor. In such scheme, once the toxin is bound, the stability of the complex only depends on the free energy of the CTX/channel interaction surface, and on the internal potassium concentration. Thus, the external K$^+$ sensitivity of $k_{off}$ can be explained if the selectivity filter 'knows' what the external K$^+$ concentration is in the toxin bound state.

## The toxin-channel complex has no memory of the ionic composition

Is it possible that the pore *remembers* the identity of the ions that were present in the external solution at the instant of the blockade event? *Figure 4* shows that, after removal of CTX from the solution, the dissociation rate was always sensitive to the identity of the main external cation present during the perfusion. This lack of memory is consistent with the existence of a communicating pathway between the selectivity filter and the external solution in the CTX-blocked channel.

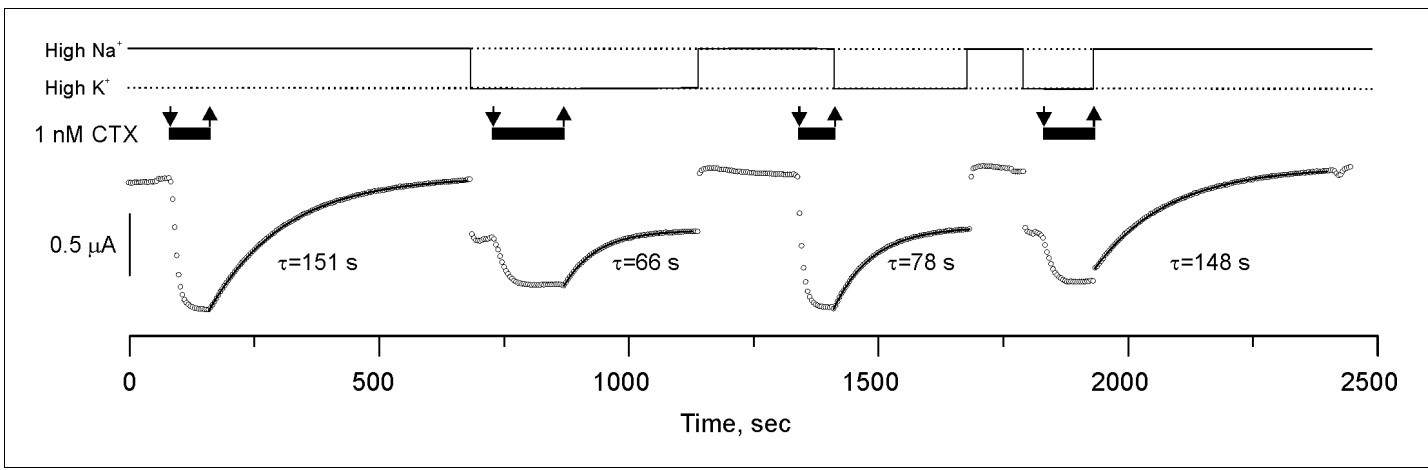

**Figure 4.** CTX blockade of Shaker-F425G has no memory. Potassium currents from an oocyte perfused with solutions having 1 nM CTX (black bars on top of the traces) in alternating High K$^+$ or High Na$^+$ conditions (continuous line on top of the figure). The sudden changes of current amplitude correspond to the changes in the potassium driving force produced by the High K$^+$/Low K$^+$ solution exchange. Toxin application occurred in these two recording solutions and recovery was also in either High K$^+$ or Low K$^+$. Toxin dissociation time courses were fit to a single exponential function (the time constants are below each curve). Note that the rate of recovery was solely specified by the ionic composition of the solution being perfused, indicating that binding retains no memory of the ionic composition at the instant of blockade. We performed three more high-Na/high-K experiments with similar results (see data repository).

DOI: https://doi.org/10.7554/eLife.46170.005

## The trans-toxin sensitivity is not coupled to voltage dependency in the closed channel

As discussed previously, depolarization may favor K$^+$ occupancy at the proximal end of the pore, electrostatically weakening the toxin-channel complex (*MacKinnon and Miller, 1988*; *Park and Miller, 1992*; *Goldstein and Miller, 1993*; *Ranganathan et al., 1996*). However, in the occluded channel pore, potassium occupancy should not be voltage dependent because the pore should be mostly isopotential. We reasoned that both phenomena, voltage dependency and trans toxin sensitivity, could occur if the toxin transiently clears the external side, thus, the electric field could be reestablished across the pore, communicating the selectivity filter with the external solution.

To our knowledge, voltage and external-K$^+$ sensitivity, together and by separate, have not been tested in CTX/Shaker- equilibrium. Because CTX dissociation rate from Shaker-F425G, our background construct, is in the order of several hundreds of seconds (*Figures 2–4*), we chose not to use it to study its voltage dependency. Instead, we looked for a conservative experimental condition in which we could separate voltage-dependence from K$^+$-sensitivity. Affinity of CTX for wild-type Shaker is 2000-fold lower than for Shaker-F425G, but very sensitive to ionic strength. Thus, by controlling the ionic strength, we could fit the toxin sensitivity to our experimental design. We used 5 nM CTX in wild-type Shaker in reduced ionic strength (~15% of normal; see Materials and methods) to promote a significant blockade that, also evidences its voltage sensitivity (*Figure 5A–B*). K$^+$-current traces in 10 mM NaCl (left, 10-Na$^+$) or 10 mM KCl (right; 10-K$^+$) in the bath show a marked inhibition by CTX (See Materials and methods). Channel activation appears kinetically delayed and is sensitive to the main external cation; the stronger inhibition is seen in 10-K$^+$, and larger distortion in 10-Na$^+$. These distortions are reminiscent of the blockade of κ-conotoxin-PVIIA (κ-PVIIA), a pore blocking peptide-toxin, on the Shaker K-channel (*Scanlon et al., 1997*; *García et al., 1999*; *Terlau et al., 1999*; *Naranjo, 2002*). κ-PVIIA appears to delay activation, but such delay is in fact a voltage dependent toxin-binding relaxation occurring after the channels were opened by the voltage pulse. Point-by-point ratios between current traces with toxin and their respective traces without toxin, allow for a better description of these voltage dependent relaxations (*Figure 5D*; see Materials and methods). The traces depicting the currents ratio start from a similar inhibition value near ~0.3, which represents the fraction of channels inhibited at resting, and exponentially grow to reveal increasingly weaker inhibition at more positive voltages. For kinetic analysis, we fitted single exponential functions to these relaxations (Blue traces in *Figure 5D*). *Figure 5E* plots the time constants (τ, black symbols, left axis coordinate) and the asymptotic inhibition (($I_{Tx}/I_{Con})_{inf}$, open symbols, right axis coordinate) for five different experiments for each external solution. For a 1:1 stoichiometry, we obtain (*MacKinnon and Miller, 1988*; *Goldstein and Miller, 1993*; *García et al., 1999*; *Terlau et al., 1999*):

$$\tau = \frac{1}{k_{on}[\text{CTX}] + k_{off}} \tag{3a}$$

$$\left(\frac{I_{Tx}}{I_{Con}}\right)_{inf} = \frac{k_{off}}{k_{on}[\text{CTX}] + k_{off}} \tag{3b}$$

*Equations (3a)* and *Equations (3b)* form an equation system with two-unknowns ($k_{on}$ and $k_{off}$) that we solved for each voltage. *Figure 5F* show results of $k_{on}$ (filled symbols), and $k_{off}$ (open symbols) in 10-Na$^+$ and 10-K$^+$ conditions. Interestingly, in comparison with normal high potassium solutions, $k_{on}$ grew by ~40 fold in 10-K$^+$, while $k_{on}$ grew only ~10 fold by going from high sodium solution to 10-Na$^+$ (*Goldstein and Miller, 1993*). Such differential sensitivity is expected because K$^+$ competes directly with CTX, while Na$^+$ possibly does not. Voltage dependency resides mostly in $k_{off}$, while $k_{on}$ is nearly voltage independent as seen before for κ-PVIIA and other α-KTX pore-occluding toxins (*Anderson et al., 1988*; *MacKinnon and Miller, 1988*; *Goldstein and Miller, 1993*; *García et al., 1999*; *Terlau et al., 1999*). As for κ-PVIIA and CTX in other experimental conditions, $k_{off}$ grows $e$-fold every ~50 mV, which corresponds to an effective electrical valence, $z$ ~0.5 (*Goldstein and Miller, 1993*; *García et al., 1999*; *Naranjo, 2002*). As with the experiments in normal ionic composition (*Figures 2–4*), $k_{off}$ is also 2-fold larger in 10-K$^+$ than in 10-Na$^+$ solutions, showing that in these conditions, with ~300 fold faster kinetics, $k_{off}$ retains trans-toxin ion sensitivity.

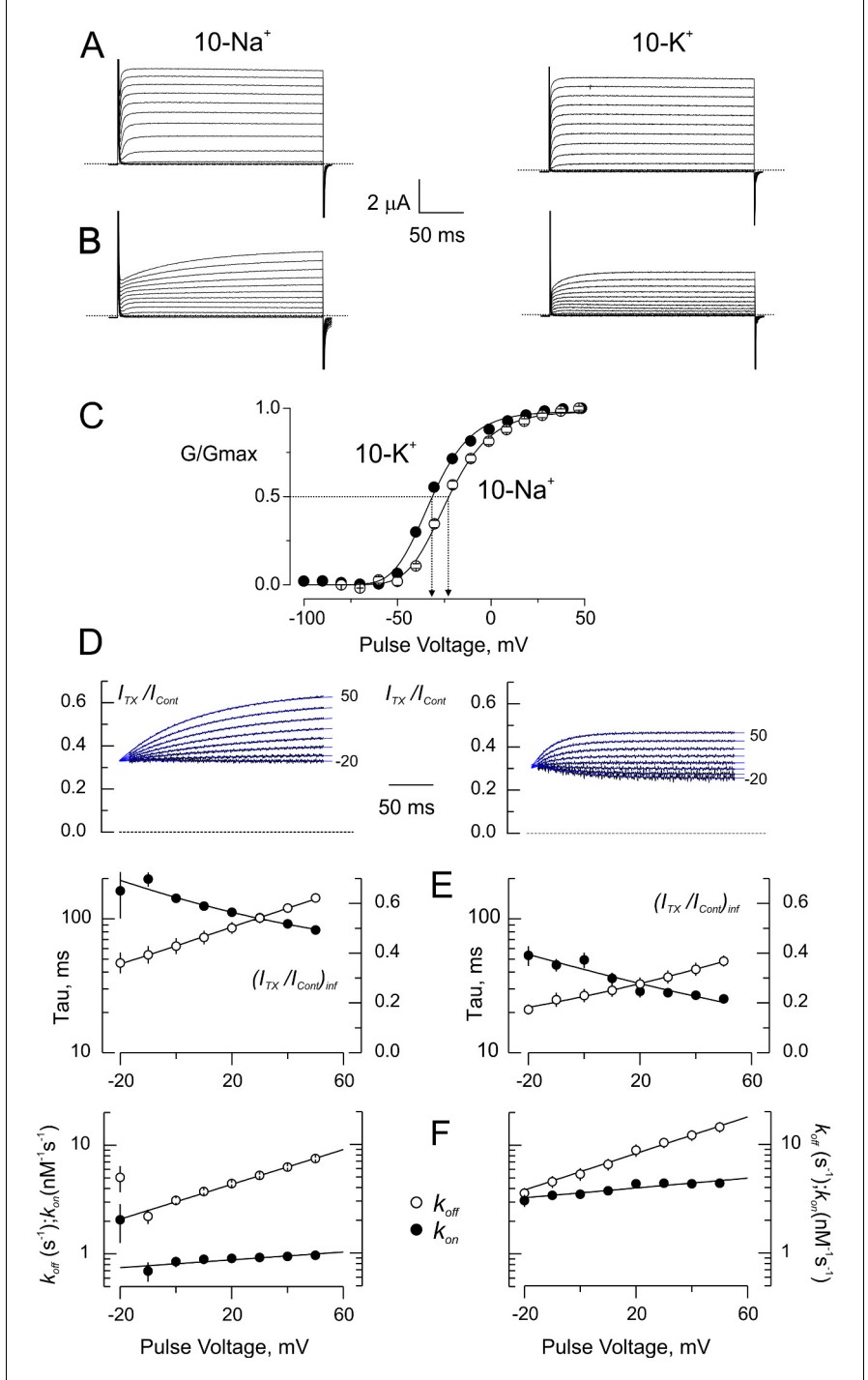

**Figure 5.** Blockade of wild type-Shaker by CTX in low ionic strength. (**A**) Potassium currents from the same oocyte recorded in external solutions in which the main cation is ~ 10 mM Na$^+$ (10 Na$^+$; left) or ~ 10 mM K$^+$ (10 K$^+$; left). (**B**) Same oocytes as in A, in the presence of 5 nM CTX. The time course of the currents appears as a delayed channel activation (**C**). Relative conductance (G/Gmax) vs. Voltage in low ionic strength. Data points are average of 5 different experiments in 10-Na$^+$ (open circles) and 10-K$^+$ (filled circles). The solid lines are fits to a Boltzmann distribution. The dashed lines indicate the approximate voltages at which G/Gmax is 0.5 (−30 and −22 mV, for 10-K$^+$ and 10-Na$^+$, respectively). (**D**) Point-by-point quotient ratios of the CTX current traces divided by the control current traces at their corresponding voltages. The blue lines are mono-exponential fits to the ratios. Note that all relaxations converge to the same point at the beginning of the pulse, pointing to the CTX inhibition at resting. (**E**) Fitting parameters of the traces in C. The average time constants (filled symbols) and the asymptotic, or
*Figure 5 continued on next page*

*Figure 5 continued*

equilibrium, inhibition at each voltage (open symbols) from five experiments each were used to calculate the association and dissociation rates. Lines have no theoretical meaning. (**F**) Rates constants of CTX binding to the channel calculated according to *Equation 3a and 3b*. The continuous lines are mono-exponential fits of the function $k = k_{(V=0)}e^{\frac{z\delta FV}{RT}}$, where $F$, $R$ and $T$ have their usual meaning. $V$ is the applied voltage and $z\delta$ is the effective valence of the voltage dependence. For five measurements in each condition, the mean ($\pm$ SEM) were: in 10 Na$^+$, $k_{off(V=0)}$=3.02 $\pm$ 0.02 s$^{-1}$ with $z\delta$=0.47 $\pm$ 0.02; $k_{on(V=0)}$=812 $\pm$ 26 µM$^{-1}$s$^{-1}$ with $z\delta$=0.106 $\pm$ 0.026; meanwhile in 10 K$^+$, $k_{off(V=0)}$=5.6 $\pm$ 0.16 s$^{-1}$ with $z\delta$=0.4 $\pm$ 0.016 and $k_{on(V=0)}$=3572 $\pm$ 96 µM$^{-1}$s$^{-1}$ with $z\delta$=0.133 $\pm$ 0.023.
DOI: https://doi.org/10.7554/eLife.46170.006

Is the voltage dependency seen in open blocked-channel due to the drop, albeit transiently, of the electric field across the pore? As in the open channel, the closed-blocked channel should retain trans-toxin sensitivity, but in contrast, the pore should be *permanently* isopotential since the electric field drops across the intracellular activation gate. To test this prediction, we measured the level of tonic inhibition, in closed channels, as a function of the holding potential (*Figure 6*). *Figure 6A–B* show comparison of current traces in 10-K$^+$ from the same oocyte, with or without toxin, elicited from holding voltages of $-120$ mV (Left) and $-60$ mV (right). Currents activated from a holding voltage of $-60$ mV are ~10% smaller due to slow inactivation. We have shown in the past that slow inactivation and CTX and κ-PVIIA binding are mutually insensitive, then, we did not expect additional effects other than current reduction by blockade (*Naranjo, 2002*; *Oliva et al., 2005*). Also, in agreement with this idea, the structure of the CTX/Kv-complex does not show any indication of toxin induced conformational effects in the pore (*Banerjee et al., 2013*). The representative results in *Figure 6C* are consistent with this idea because they show that the kinetic parameters of open channel inhibition relaxation are independent from the level of resting inhibition at holding voltages of $-60$ or $-120$ mV. Thus, the time constants and asymptotic inhibition at positive voltages were identical upon opening (*Figure 6D*). *Figure 6E*, shows a summary plot of steady state inhibition ($I/Io$) as function of the voltage for 10-Na$^+$ and 10-K$^+$ data (mean $\pm$ SEM of 5 and 6 oocytes, respectively). This graph is divided in two areas, for voltages $\leq -60$ mV, we plot the resting inhibition; for voltages $\geq -30$ mV, we plot the asymptotic inhibition obtained from *Figure 6D*. Both datasets showed a complex voltage-dependency for $I/Io$: at voltages $< -100$ mV it is voltage independent, biphasic at voltages $>-100$ mV, and with a positive upstroke for voltages $\geq -30$ mV (see below). Thus, the toxin blockade is voltage dependent in open channels while is voltage-insensitive at voltages $<-90$ mV where the channel´s apparent open probability ($P_O$) should be below $10^{-6}$ (*Islas and Sigworth, 1999*; *González-Pérez et al., 2010*; *Ishida et al., 2015*).

To address the different blockade regimes, the following simplistic equilibrium describes binding to closed and open channels (*Scheme 1*):

$$
\begin{array}{ccc}
 & C \overset{K_{DC}}{\longleftrightarrow} C \cdot Tx & \\
K_{V1} \uparrow\downarrow & & \downarrow\uparrow \; K_{V2} \\
 & O \underset{K_{DO}}{\longleftrightarrow} O \cdot Tx &
\end{array}
$$

**Scheme 1.** CTX binding equilibrium to open and closed channels.
DOI: https://doi.org/10.7554/eLife.46170.008

where $K_{DC}$ and $K_{DO}$ are the toxin dissociation constants for the closed and open state respectively. $K_{V1}$ and $K_{V2}$ are the open-close voltage dependent equilibrium constants of the toxin-free and toxin bound channels, respectively. All quantities, except $K_{V2}$, are easily accessible: $K_{V1}$ can be obtained from the conductance vs. voltage relationship; $K_{DC}$ from the binding equilibrium at very negative voltages; $K_{DO}$ from the toxin binding to open channels, while $K_{V2}$ could be addressed by assuming microscopic reversibility, ie:

$$K_{V2} = \frac{K_{DC}}{K_{DO}} K_{V1}$$

The inhibition scheme predicts that the fraction of toxin-free channels, $U/U_{max}$, is:

$$\frac{U}{U_{max}} = \frac{1 + K_{V1}}{1 + K_{V1} + [\text{Tx}]\left(\frac{1}{K_{DC}} + \frac{K_{V1}}{K_{DO}}\right)} \tag{4}$$

Making $K_{V1} = \frac{P_o}{1 - P_o}$ in **Equation 4** we obtain a more intuitive expression for the fraction of toxin-free channels:

$$\frac{U}{U_{max}} = \frac{1}{1 + \frac{[\text{Tx}]}{K_{DO}}P_o + \frac{[\text{Tx}]}{K_{DC}}(1 - P_o)} \tag{5}$$

This equation describes two well defined inhibition regimes: a voltage independent one at $P_O \sim 0$ and a monotonically voltage dependent at $P_O \sim 1$. Thus, at large negative voltages $P_O$ is small, then:

$$\frac{U}{U_{max}} \sim \frac{1}{1 + \frac{[\text{Tx}]}{K_{DC}}}$$

$U/U_{max}$ at resting is voltage independent. In contrast, if $P_O$ is $\sim 1$, the binding equilibrium should be dominated by de voltage dependency of $K_{DO}$

$$\frac{U}{U_{max}} \sim \frac{1}{1 + \frac{[\text{Tx}]}{K_{DO}}}$$

If we assume that $I/Io$ is a good estimation of $U/U_{max}$, the biphasic inhibition results from $P_O$ *growing* with the voltage (**Figure 6E**). The solid lines in **Figure 6E**, are fits to **Equation 5**. In resting channels ($P_O \sim 0$) the binding (or $I/Io$) is independent of the holding voltage (Vh in **Figure 6**), while when the $P_O \sim 1$, it is fully voltage dependent. This behavior has been reported for CTX binding to BK-channels and for κ-PVIIA binding to Shaker K-channels (**Anderson et al., 1988**; **García et al., 1999**; **Terlau et al., 1999**). Thus, in the open channel the toxin-binding equilibrium is voltage and potassium dependent, meanwhile, in closed channels (with the activation gate blocking the pore), the toxin-binding remains sensitive to the external potassium but is voltage independent.

## Voltage and trans-toxin sensitivities reveal a transient pore-communication

The effective electrical valence, *z*, of the voltage dependence of CTX binding to BK channels is $\sim 1$ $e_o$, which vanishes entirely upon removal of the internal potassium (**MacKinnon and Miller, 1988**; **Park and Miller, 1992**). Meanwhile, in Shaker $z = 0.4$ $e_o$. Only half of this $z = 0.4$ $e_o$ disappears after total removal of the internal potassium (**Goldstein and Miller, 1993**). A similar internal potassium sensitivity ($z \sim 0.25$ $e_o$) is seen in κ-PVIIA dissociation rate (**García et al., 1999**). It is interesting that the amplitude of the $K^+$-dependent *z* match reasonably well with the fraction of the electric field that drops across the selectivity filter in open-conducting BK and Shaker channels (50–100% vs. 8–15%, respectively) (**Díaz-Franulic et al., 2015**).

Voltage sensitivity of toxin unbinding may reveal a transitory state in which the pore is briefly communicated with the external medium, momentarily equilibrating the selectivity filter ion composition with the external solution and, restoring the electric field across the pore. Thus, the toxin final dissociation rate is voltage- and external ion-sensitive because the toxin wobbles in its binding site, forming transition state(s) resembling the unblocked channel. We propose here that such transition states go undetected because of sub μs life-time duration (see below).

## Molecular Dynamics simulations show wobbling

CTX covers $\sim 400$ Å$^2$ of the external channel vestibule, and the protein–protein interface shows 6–10 well-characterized interacting partners, being one of them the pair formed by CTX-Lys-27 ε-amino group with Shaker's Tyr-445 carbonyls via $K^+$-ions, in the S1 binding site of the selectivity filter (**Ranganathan et al., 1996**; **Banerjee et al., 2013**). If the toxin is wobbling in its binding site, the interacting partners would be in dynamic equilibria, establishing a lively network of forming and vanishing contacts. Hence, MD simulations could describe at an atomic level the multiple contacts dynamic between CTX and the Kv-channel outer face. Thus, MD systems were built by embedding the structures of the channel-toxin (PDB:4JTA) or the channel alone (PDB:2R9R) in a phospholipid

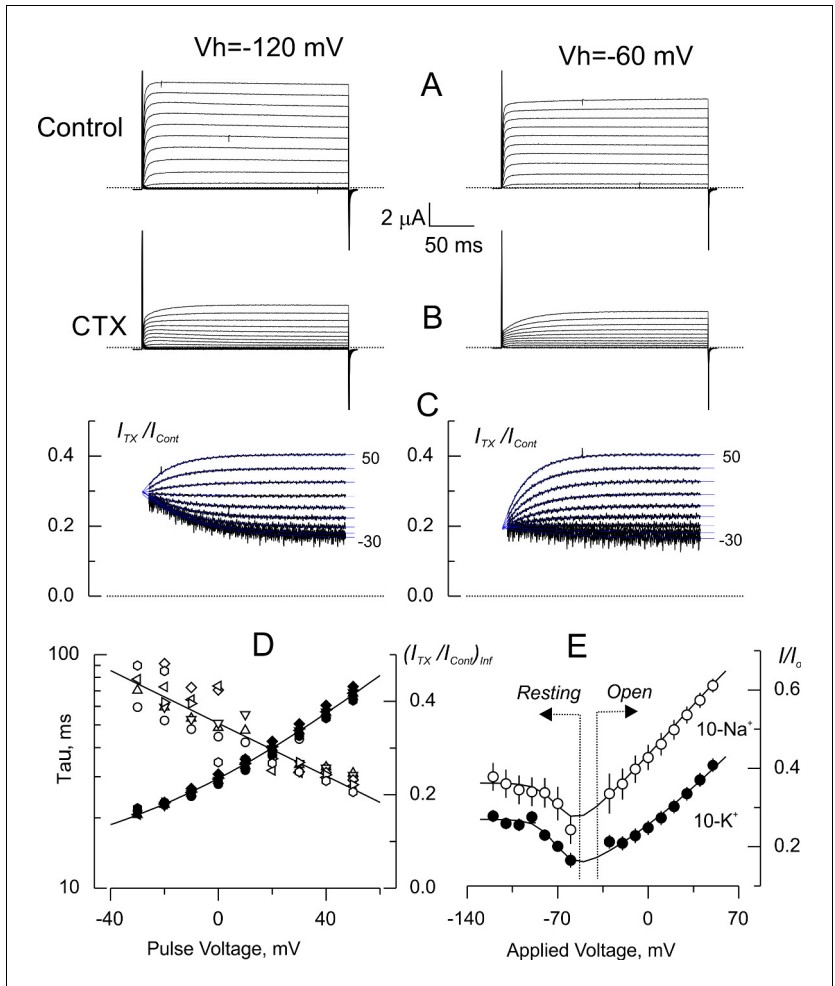

**Figure 6.** Voltage dependence of tonic CTX inhibition of wild type-Shaker. (A) Representative recordings of potassium currents elicited from holding voltages ($Vh$) of −120 mV (left) and −60 mV (right) in 10 K$^+$ solution in the same oocyte. The small reduction of currents is probably due to Shaker slow inactivation (**González-Pérez et al., 2008**). (B). Potassium currents elicited in the presence of 5 nM CTX. Same oocyte and $Vh$ as in A. (C) Point-by-point quotient ratios of traces as in **Figure 5**. The blue traces are mono-exponential fits to the resulting relaxations within the length of the voltage pulse. Note that the CTX inhibition at resting is different between $Vh$= −120 and $Vh$= −60 mV. (D) Fitting parameters of the traces in C. The average time constants (open symbols) and equilibrium inhibition at each voltage (filled symbols) from relaxations elicited from different $Vh$ in the same oocyte (circles: −120 mV; triangles: −120 mV; inverted triangles: −100 mV; diamonds: −90 mV; left rotated triangles: −80 mV; right rotated triangles: −70 mV; hexagons: −60 mV). The continuous lines are polynomial fits with no theoretical meaning. (E). Tonic inhibition as a function of the voltage. *Resting* indicates measurements made from the resting inhibition at different $Vh$, while *Open* specifies the inhibition at equilibrium from the asymptotic values of the voltage dependent relaxations as those shown in C. Each data point corresponds to average of 5 and 6 different oocytes (± SEM) for 10-Na$^+$ and 10-K$^+$ data, respectively. The continuous lines are fits to **Equation 5** with the Levenberg-Marquardt method. To fit **Equation 5** we used: $K_{DO} = K_{DO(V=0)}e^{\frac{zFV}{RT}}$ where $z$ is the toxin effective valence, and $Po = \frac{1}{1+e^{\frac{-ZF(V-Vo)}{RT}}}$ where $Z$ is the effective valence of Kv-channel opening, $Vo$ is the voltage at which $Po$ = 0.5. The rest of the parameters as declared. The fit parameters (± SE) for 10-Na$^+$ were: $K_{DO}$ = 3.9 ± 0.16 nM, $z$ = 0.36 ± 0.03, $Vo$ = −67 ± 7 mV, $Z$ = 4.0 ± 3.1, and $K_{DC}$ = 2.81 ± 0.27 nM; for 10-K$^+$ were: $K_{DO}$ = 1.7 ± 0.05 nM, $z$ = 0.29 ± 0.02, $Vo$ = −66 ± 2 mV, $Z$ = 3.9 ± 0.89 and $K_{DC}$ = 1.84 ± 0.06 nM.

DOI: https://doi.org/10.7554/eLife.46170.007

bilayer (POPC) separating two aqueous compartments loaded with the equivalent to 100 mM KCl solutions. After 60 ns of non-restrained equilibration simulation, an electrical potential was applied at the cytosolic side. We applied +100 mV, in one simulation, or +500 mV (in two simulation

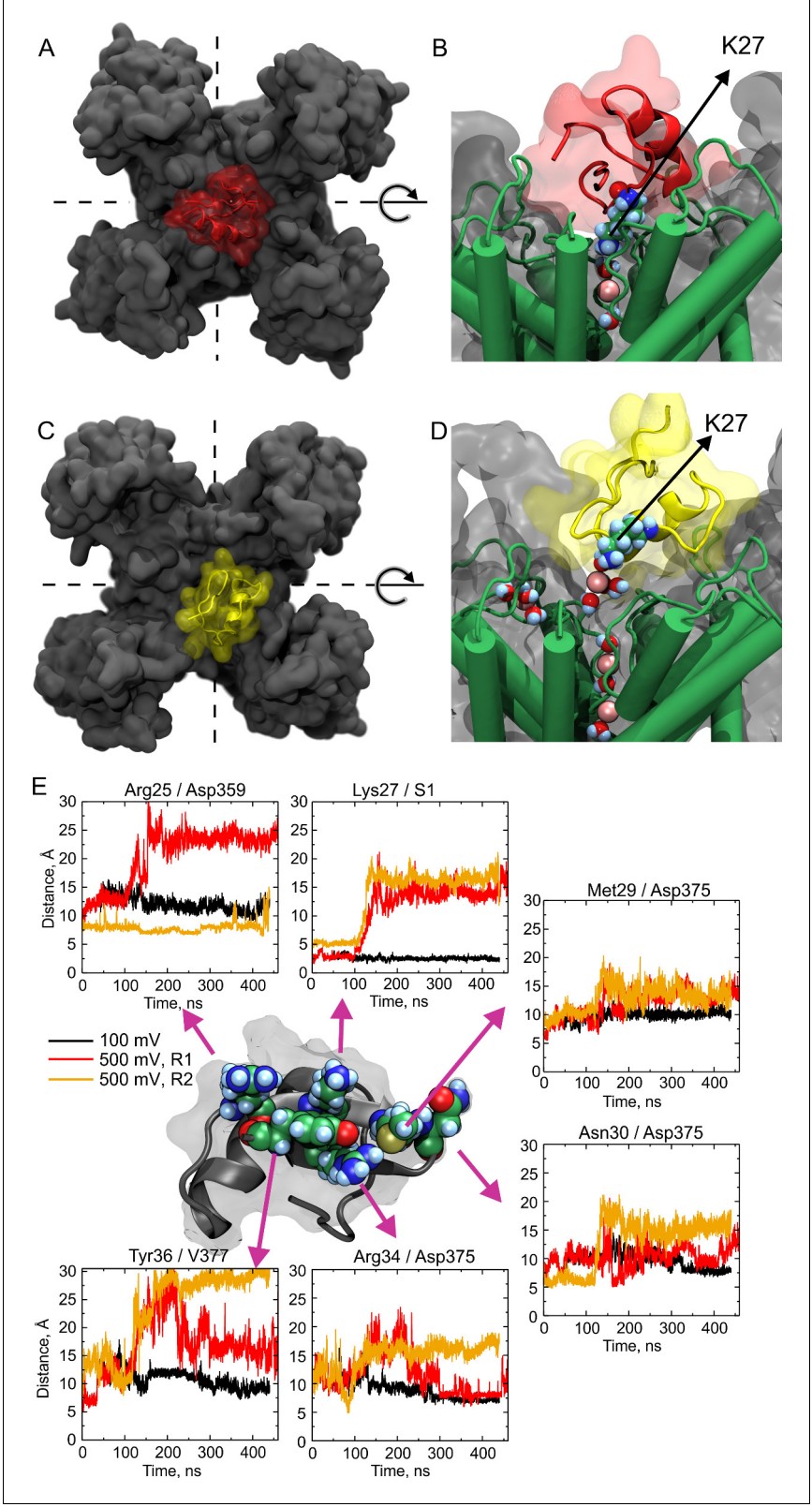

**Figure 7.** Molecular dynamics show large wobbling. (**A, B**) CTX bound to the Kv-channel at the beginning of Replica 2. The toxin is attached to the vestibule and the ε-amino group of Lys27 sits in S1 replacing permeant ions in the site. (**C, D**) CTX bound to the Kv-channel 400 ns later. In **C**, CTX end up ~ 90° rotated CCW with respect to A while Lys27 amino group is elevated ~10 Å from the floor of the interaction surface; K⁺ ions populate the

*Figure 7 continued on next page*

*Figure 7 continued*

interaction surface and fill the selectivity filter vacant S1, increasing the pore occupancy. (E) Distances between sidechain contacts at the bottom of the interaction surface. The plotted values are the distances between the centers of mass of the indicated interacting sidechains, except for S1, in which the center of the site was used. CTX´s Arg25, Lys27, Met29, Asn30, Arg34, and Tyr36 are depicted as reference at the center of the figure. The Kv-channel residues were: Asp359 from chain D, Asp375 from chain B for Met29 and Asn30, and from chain A for Arg34. Val377 is from chain A. A movie showing wobbling and potassium competition with with Lys27 amino group rebinding is available on Dryad (https://doi.org/10.5061/dryad.0p77qk4).
DOI: https://doi.org/10.7554/eLife.46170.009

The following figure supplements are available for figure 7:

**Figure supplement 1.** CTX binding alters the electrostatic profile along the pore.
DOI: https://doi.org/10.7554/eLife.46170.010

**Figure supplement 2.** Electrostatic profiles along the pore of the Kv-channel with CTX bound in tight mode.
DOI: https://doi.org/10.7554/eLife.46170.011

---

replicas). *Figure 7* shows representative snapshots of the CTX bound to the Kv-channel at the beginning of the simulation (*Figure 7A,B*) and after 400 ns (*Figure 7C,D*). Note that in the later stage, the toxin is rotated about ~90° counter-clockwise (*Figure 7C*) and is raised ~10 Å from the channel surface (*Figure 7D*). Therefore, while Lys27 amino group is detached, ions *sneak-in* between both surfaces, and K⁺-ions occupy the S1 site in the selectivity filter, recovering the characteristic double occupancy of the unblocked channel. *Figure 7E* depicts the distance separating interacting sidechain-pairs, at the floor of the interaction surface as function of simulation time: CTX-Arg25/Kv-Asp359, CTX-Lys27/Kv-S1 (Tyr373), CTX-Met29/Kv-Asp375, CTX-Asn30/Kv-Asp375, CTX-Arg34/Kv-Asp375, and CTX-Tyr36/Kv-Val377 (in Shaker numeration they correspond to Asp431, Tyr445, Asp447, and Met449, respectively). Each panel shows the distance between the center of mass of each sidechain of the pair-members as a function of time, for the three MD simulations: at +500 mV (replicas R1 and R2, traced in red and orange, respectively) and one at +100 mV (black). While along the simulation at +100 mV no pair became separated >10 Å, replicas R1 and R2 show significant pair-separations (10–20 Å) beginning around 60–100 ns. These concurrent departures report partial toxin detachment from its binding site. However, some contact dynamics show some independence. For, example, in R1 (red traces), Arg25, Lys27, Arg34 and Tyr36 separate by ~10 Å, meanwhile Met29 and Asn30 remained close to their respective interacting points; but, while Arg25 and Lys27 remained separated, Arg34 and Tyr36 returned to baseline around 240 ns into the simulation. Remarkably, R2 (yellow traces) shows that while Tyr36 and Lys27 acquire separation of ~20 and ~10 Å from their cognates, respectively, their physically flanking residues, Arg25 and Met29 remained attached to the channel. Thus, CTX seems to wobble as a rigid body, with residue-residue correlated movements; however, the dynamics of individual neighboring contact points have large degrees of autonomy. Considering the crudity of our analysis, this is an outstanding result that provides support to the idea that toxin-wobbling may uncover the ion conduction path when Lys27 separates from S1-site, while at the same time, neighboring sidechains remain attached to their contacts.

## Is the trans-pore electric field affected during the wobbling?

We indirectly addressed this question by calculating the electrostatic potential along the pore in both, the native and toxin-bound channel structures. As with the above-mentioned Replicas, after the 60 ns equilibration, an external +500 mV electric potential was applied at the intracellular side of the native Kv-structure. We calculated the electrostatic potential along the pore; collecting averages every 100 ns non-overlapping windows of simulation time. According to *Figure 7*, the first 100 ns should represent the electrostatic profile of the occluded channel by a tightly bound toxin (See *Figure 7—figure supplement 1* blue trace, and *Figure 7—figure supplement 2*, red and yellow traces). This idea was corroborated by the similitude to another 100 ns-averaged electrostatic profile in a system in which the toxin was kept tightly bound by the application of harmonic restrains to 11 crystallographic distances in the PDB file separating contact points between the toxin and the channel, including Lys27 to S1 (*Figure 7—figure supplements 1* and *2*). In Replica 2, in the ensuing 100-ns averages Lys27 was conspicuously detached from the pore. In these later averages, the electrostatic potential profiles look very similar to the toxin-free channels (see *Figure 7—figure*

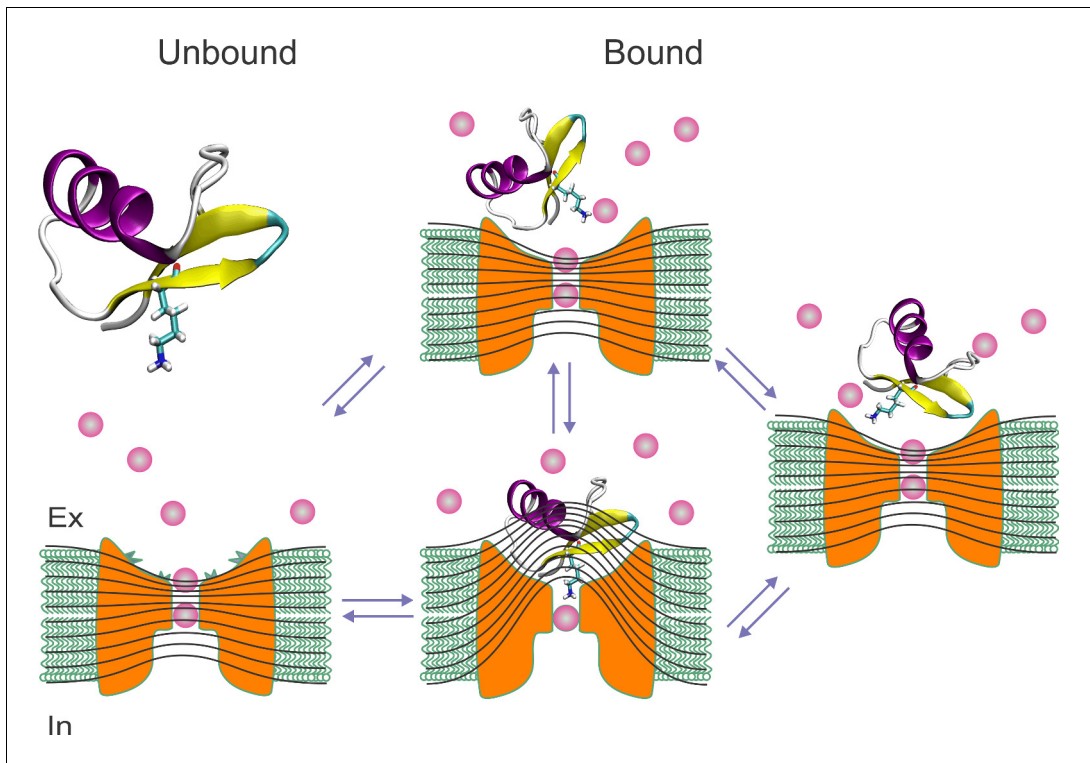

**Figure 8.** Toxin wobbling restores the concentration and the electric gradient across the pore. (**A**) The bound toxin transits among different contact configurations in quasi-equilibrium. Some configurations restore the pore occupancy by permeant ions, and the trans-pore electric field, as in the unblocked channel, and may yield to the final dissociation step.

DOI: https://doi.org/10.7554/eLife.46170.012

supplement 1). Thus, the electrostatic profile along the pore of the wobbling toxin-bound channel resembles that of the unblocked channel. Such similarity could be the result of the reestablishment of ion-occupancy.

## Unbinding dynamics

Macroscopically speaking, toxin unbinding is a first order phenomenon like a radioactive decay. Thus, the dissociation rate should be:

$$k_{off} = A_o \times e^{\frac{-\Delta G^{\ddagger}}{RT}} \tag{6}$$

where $A_o$ is the maximal possible rate and $\Delta G^{\ddagger}$ is the activation energy (>0 by definition), defined here as the energy difference between the toxin-channel complex and the last transition state (‡) from which the toxin dissociates. The access to the last transition state is an event with a very low probability and in our case occurs every 0.1-180 seconds in average. These are very extended periods for molecular interaction, during which many intermediate transitions may occur before unbinding. One or few of these intermediates would lead to the final toxin unbinding, while others may only expose sections of the interaction surface to the external solution, without ending the bound state. Some intermediates would expose the selectivity filter, restoring the electric field and ionic communication with the external solution. Thus, while the toxin is bound to the channel, multiple contact configurations would interconvert in the non-conducting toxin-bound state. As long as their exchange kinetics is much faster than the toxin-channel dissociation rate, these multiple conformations can be treated as if there were in equilibrium (*Dill and Bromberg, 2011*).

Let us assume that in the toxin-channel interacting surfaces there are several partners or contact points (*n*) in binding-unbinding equilibrium, thus the multiple conformations correspond to the

ensemble of all individual association/dissociation transitions at each contact point along the interaction surface (*Figure 7A*). The lifetime of the channel-toxin complex ends when all individual contact points dissociate. Thus, we can define the macroscopic dissociation rate as:

$$k_{off} = A_o \times K^{\ddagger}$$

where $K^{\ddagger}$ is the compound dissociation constant involving all individual point-equilibria (or the probability of the final transition state). Because the neighboring contact trajectory in our MD simulations show large degrees of independence, but mostly for simplicity, let us assume that the $n$ contacts forming the interaction surface are mutually independent and energy-additive, then: $\Delta G^{\ddagger} = \Delta G_1 + \Delta G_2 + \ldots + \Delta G_{n-1} + \Delta G_n$. Thus, $k_{off}$ is proportional to the probability that all contact points are dissociated

$$k_{off} = A_o \times K_1 K_2 \ldots K_j e^{\frac{zFV}{RT}} K_{n-1} K_n \tag{7}$$

where $K_j$ represents the voltage dependent equilibrium for the specific interaction between CTX-Lys27 side-chain and the selectivity filter (*MacKinnon and Miller, 1988*; *Park and Miller, 1992*; *Goldstein and Miller, 1993*). Because $k_{off}$ grows with the voltage, $z$ is positive. Thus, the overall dissociation rate constant should be:

$$k_{off} = A_o \times K_o^{\ddagger} e^{\frac{zFV}{RT}}$$

where, at $V=0$:

$$K_o^{\ddagger} = \prod_{i=1}^{n} K_i$$

There is a dynamic interaction between Lys-27 sidechain and the selectivity filter, such that, just few nanoseconds after the ε-amino group dissociates, the open channel electric field and proximal pore occupancy get restored. Our two +500 mV MD simulations show that briefly after the application of the transmembrane electric field, Lys27 detaches from S1 (*Figure 7E*), an event that did not happened in the +100 mV simulation. Consistent with the original proposal for voltage and ion enhanced dissociation, these two simulations suggest that the electric field, or permeant ions, push away the side-chain amino group from its binding site, (*MacKinnon and Miller, 1988*; *Park and Miller, 1992*; *Swartz, 2013*). Nevertheless, these results are too preliminary to be conclusive. Alternatively, externally located permeant ions gaining full access to the selectivity filter would electrostatically slow the rebinding of Lys-27 sidechain in a mostly restored electric field. Then, the voltage dependency would reside in the re-binding of Lys27 ($k_{jon}$) in the $j$-contact equilibrium ($K_j$). Then:

$$K_j e^{\frac{z_j FV}{RT}} = \frac{k_{joff}}{k_{jon} e^{\frac{-z_j FV}{RT}}} \tag{8}$$

where $k_{joff}$ is the local dissociation rate. The voltage dependency of rebinding would be larger for larger conductance K-channels because a larger fraction of the electric field drops across the selectivity filter (*Díaz-Franulic et al., 2015*). Thus, according to *Equations 7 and 8*, the macroscopic dissociation rate, $k_{off}$, will be voltage dependent because Lys27 re-binding would be voltage dependent. Likewise, the occupancy of the S1 site by permeant ions should compete with Lys27 re-binding, thus, $k_{off}$ enhancement in *Equation 2a* describes how $K_j$ grows with the external $K^+$. Since the macroscopic association rate is diffusion limited and then, voltage independent (*Figures 5*, *6*), stating that $k_{jon}$ is voltage dependent could be controversial. But, the re-binding of Lys27 (or any other contacting residue) would not be diffusion-limited because the toxin is already bound. In fact, voltage dependent association is common for blockers as TEA in which binding is not rate limiting (*Thompson and Begenisich, 2003*).

An inference from *Equation 8* is that toxin association and Lys27 re-binding both compete with $K^+$ ions over the S1 site. Then, a comparison between $K^+$ dependent enhancement of $k_{off}$ and $K^+$ dependent decrease of $k_{on}$ (*Equations 2a and 2b*) should account for the differences in apparent affinity for $K^+$ of the S1site between toxin-free and toxin occupied channels. Thus, assuming that S1

does not bind $Na^+$ and is available when CTX wobbles, the ratio $K_{K2} / K_{K1} \sim 0.004$ provides a lower limit for the fractional time that S1 is externally $K^+$ available in the CTX-blocked channel.

Of course, the unbinding dynamics by wobbling proposed here is an oversimplification for two rigid binding structures. Peptide toxins, having cystine knots, and the external vestibule of Kv-channels seem to behave as rigid bodies that retain their structure upon binding (*Banerjee et al., 2013*). Therefore, dynamics of neighboring contacts in the interaction surface should show mutual correlation. Nevertheless, despite this large oversimplification, the assumption of local energy-additivity and independence seems to work fine for CTX/Kv-channels and other interacting protein complexes as Barnase/Barstar, since the mutant cycle analyses, based on this principle, accurately predicted the contact map in these two systems (*Hidalgo and MacKinnon, 1995*; *Schreiber and Fersht, 1995*; *Naranjo and Miller, 1996*; *Ranganathan et al., 1996*; *Frisch et al., 2001*; *Banerjee et al., 2013*). Thus, we can image several energetically similar configurations in which some of the contacts could vanish until the toxin swivel back like a hinged cap (*Figure 8*). The tightly sealed crystal structure of the CTX/Kv-channel complex possibly represents the most likely conformation, but the blocking toxin would need just a few nanoseconds in a rare permissive conformation to restore the electric field and equilibrate the pore with the external solution; going undetectable due to the time-resolution of electrophysiological recordings.

Alternatively, the toxin binding could be *leaky*, allowing a residual current. In this case, the toxin unbinding may not be voltage dependent because most of the electric field should drop permanently across the toxin body, leaving the pore always isopotential. There are instances of incomplete pore blocking in $Na^+$-channels by μO-conotoxins that leave measurable residual currents (*French et al., 1996*). Although, voltage dependency has not been tested in these toxins, we would not expect enhanced dissociation by permeant ions because they do not plug the pore. On the other hand, blockade by α-KTX and κ-PVIIA appears to be complete. Residual currents, if exist, should be well below $10^{-3}$ of the conducting channels (*Anderson et al., 1988*; *Aggarwal and MacKinnon, 1996*; *Gross and MacKinnon, 1996*; *Naranjo, 2002*).

It is surprising that the toxin-binding equilibrium is similarly sensitive to the external ions in the closed and in the open state (*Figure 6*). When the intracellular gate is closed, once the toxin-wobbling uncovers the selectivity filter, ions in the pore would be able to equilibrate with the external solution. Upon re-blocking, the pore would somehow *retain* memory of the external ionic composition until the channel opens again. In contrast, in the open channels, such memory would not exist because the pore could rapidly equilibrate with the internal solution. Thus, the external ion effects on the toxin-binding stability could be quite different between open and closed channels. This state independence could result if the external aspects of pore quickly attain equilibrium while distal sectors take longer to, or never, do so. Thus, as *Figures 7* and *8* suggest, only the vicinity of S1 is able to equilibrate. This mechanism could explain patch-clamp experiments in which κ-PVIIA dissociates 8-fold faster from open Shaker channels in a high $K^+$ external solution than in a high $Rb^+$ one, when $Rb^+$ is the principal internal cation (*Boccaccio et al., 2004*).

## Unresolved transitions

Here we postulate that the voltage and trans-toxin sensitivity are due to the existence of at least one transient conductive state forming part of a dynamic array of contact equilibria occurring during the toxin-bound event. To restore the electric and concentration gradients, such state must be fully conductive and then, potentially detectable with electrophysiological recordings. However, single channel analysis of CTX and κ-PVIIA dwell-time blockade events in BK and Shaker, respectively, show no glimpses of conductive intervals during individual blockade events (*Anderson et al., 1988*; *Miller, 1990*; *Naranjo, 2002*). Then, the transitional state lifetime must be well-below temporal resolution of electrophysiology (let us say < 1–10 μs).

To estimate the time-scale of the wobbling dynamics we must consider how often individual contacts dissociate. Our MD simulations suggest that residues unbind few times along the 400 ns simulation time. However, our approximations are essentially crude and do not account for the individual binding dynamics. In one hand, we do not know the specific energy contribution of each contact to the global binding energy, and on the other hand, we lack a criterion to specify a distance threshold to identify local unbinding events. In *Equation 6*, let us assume that $A_o$ = 6 ps$^{-1}$ (the standard *frequency factor* from the Transition State Theory and a transmission coefficient of 1), then, the unbinding rates seen here (0.0061–3.3 s$^{-1}$) would correspond to $\Delta G^\ddagger$ of 70–85 kJ/mol (*Dill and Bromberg,*

*2011*). Thus, in average, one every $10^{13}$–$10^{15}$ attempts will effectively dissociate the toxin, unbinding all local contacts. For simplicity, let us assume that CTX makes 10 energetically identical contacts with the channel, then, the energy of each one would be ~2.8–3.5 $kT$ units. Thus, in each partial equilibrium, individual contact would be unbound ~3–6% of the time. This fractional time is enormously higher than the probability of the unbinding event, but, because we lack an explicit frequency factor for each contact, we are unable to calculate the local rates.

We must consider data in *Figure 6E* in which the toxin-blockade equilibrium switch from a voltage independent regime in the closed channel (at Vm<−90 mV), to a voltage dependent one in the open channels (Vm > 0 mV). A fit of *Equation 5* to these data yield values for the voltage dependent and independent inhibition along the open channel probability. In principle, because the trans-pore electric field can be restored in the open channel only, the transition between voltage independent to voltage dependent blockade regime should follow very closely the G-V curve (*Figure 5C*). Nevertheless, according to *Figure 6E*, the *Po*-V relationship is left-shifted by ~35–45 mV with respect to the G-V curve (*Po* = 0.5 at ~−67 mV vs. *G/Gmax* = 0.5 at −22 mV (*Figure 5C*)). Such shift is unexpected according to the microscopic reversibility used to derive *Equation 4* (ΔV ~+3 mV, for $Z$ = 3.5 $e_o$). In addition, at −67 mV, a significant amount of current should be detectable because ~50% of the channels would be open. However, the first visible current traces in *Figures 5* and *6* appear at −30 mV or higher voltages. Instead, we favor the idea that the apparent left shift in *Po* is in fact the probability that the electric field along the pore has been restored during wobbling. Then, according to the *Figure 5*, at −67 mV, when *G/Gmax* is ~$10^{-2}$, half of the channels are trans-toxin communicated. This figure is clearly an overestimation. More realistic descriptions as, for example, a Boltzmann distribution elevated to the fourth power, or current kinetic models, indicate that at −40 mV from the half activation, *G/Gmax* should be smaller than $10^{-5}$ (*Schoppa et al., 1992*; *Zagotta et al., 1994*; *González-Pérez et al., 2010*; *Díaz-Franulic et al., 2018*). Thus, a single opening longer than 10 ns, within a 10 ms blockade event, would be enough to equilibrate the pore with the electric field and the external solution in the toxin bound channel. Upon closing back, the selectivity filter would retain communication with the external side, even if the actual open probability is very small. Consequently, during the lifespan of a single channel opening (~100 μs), the toxin wobbling would be already in equilibrium and voltage dependence may emerge.

Here, we present a toxin-blocked state that is far from being rigid. In fractions of microseconds, the bound CTX visits several subsets of configuration within a network of forming and vanishing contacts. This network of interacting partners, together with thermal agitation, could yield to several different energetically equivalent dissociation pathways. There are recent and old examples of rare and ephemeral intermediate states during protein folding (*Raschke and Marqusee, 1997*; *Tapia-Rojo et al., 2019*) or during protein oligomerization (*Maity et al., 2018*). Nevertheless, remain to be shown if states that arise during mechanically stretching the protein can be spontaneously visited. Spontaneous wobbling would be similar, but much more convoluted, to the existing dissociation pathways of small ligands, as MD shows (*Cavalli et al., 2015*; *Paul et al., 2017*; *Rydzewski et al., 2018*). In fact, past MD simulations have revealed several energetically similar binding configurations in the CTX/K-channel that could interexchange (*Eriksson and Roux, 2002*; *Khabiri et al., 2011*). Thus, our findings are not entirely new. The novelty is that toxin-wobbling is in the dissociation path. The final dissociation event could occur from several different contact-configurations, or along different reaction-coordinates. This picture is seemingly at odds with simulations in which the dissociation complex is pushed or pulled along a specific space coordinate (*Khabiri et al., 2011*; *Maity et al., 2018*). Thus, wobbling could be a common feature of multi-contact protein–protein interaction. A good understanding of the molecular basis of this process could lead to develop strategies to control dissociation kinetics. Thus, for example, an intervention of the dissociation path of insulin hexamers could lead to a faster acting hormone for better, and quicker, blood-sugar control (*Zaykov et al., 2016*).

## Materials and methods

### Heterologous Expression of Shaker K-Channels

Methods employed here are described in *González-Pérez et al. (2008)*. Briefly: We used a background construct derived from Shaker H4 Δ(6–46), inserted in pBluescript SK vector commanded by

the T7 promoter. Point mutations were performed by PCR using two mutation–containing complementary synthetic oligonucleotides extended by *Pfu* polymerase over the Shaker template (Quick-Change). In addition of Shaker H4 we used Shaker F425G, a variant having a 2000-fold higher affinity for CTX (*Goldstein and Miller, 1992*) The toxin was purchased from Alomone and dissolved in recording solution in 1 mM stocks. Capped cRNAs for Shaker-IR and mutants were synthesized from a *Not I*-linearized template using MESSAGE machine (Applied Biosystems), re-suspended in 10 µl water, and stored at −80°C until use. Typically, 0.03–0.3 ngr of cRNA were injected in *Xenopus laevis* oocytes for heterologous expression following protocols and guidelines described earlier (*González-Pérez et al., 2008*).

## Electrophysiological Recordings

For two-electrode voltage clamp (TEVC) recording, oocytes were placed at the center of a 50 × 4 mm with 3 mm deep chamber in continuous perfusion. TEVC were made by either an OC-725 (Warner Instruments) or a TEV-200 amplifier (Dagan Corporation). We used a PCI-MIO-16XE-10 card (National Instruments Corp.) for voltage pulses and data acquisition under the control of the WinWCP software (University of Strathclyde, Glasgow, Scotland). The voltage and current electrode were filled with a solution of 3 M KCl, 1 mM EGTA, and 5 mM HEPES-KOH, pH 7.0, having resistances of 0.3–1.0 MΩ. Normal ionic strength recording solutions were composed of (in mM): 96 NaCl, 2 KCl, 1 $MgCl_2$, 1.2 $CaCl_2$, and HEPES-NaOH, pH 7.4 (*High Na$^+$*). For experiments to test the selectivity of the enhancement effect on toxin binding, the NaCl salt was replaced by, 96 mM of the respective chloride salt of the cation (K$^+$, Rb$^+$, Cs$^+$, NH$_4^+$). The low-ionic-strength recording solution (~0.02 M) was (in mM) 5 NaCl, 0.1 KCl, 1.8 CaCl2, 1 MgCl2, 190 Sucrose, and 10 Hepes-NaOH (*10-Na $^+$-solution*), the other recording solution contained 5 KCl instead of NaCl (*10 K*). Recording traces were exported to pClamp format (Molecular Devices) for signal analyses and curve fitting with Clampfit 10 (Molecular Devices). Data fits and plots were done with Origin 6.0 with Levenberg-Marquardt method (OriginLabs).

## Molecular Dynamics (MD) Simulations

Crystal structures of *Shaker* family of voltage-dependent channel Kv1.2 (PDB ID: 2R9R), as well as Kv1.2 channel bound to CTX (PDB ID: 4JTA), were embedded in a POPC lipid bilayer membrane and solvated with TIP3P water molecules implemented in the CHARMM force field (*Jorgensen et al., 1983*). Potassium (~0.1 M) and chloride counter ions were added to the bulk phase to mimic the experiments and to guarantee electrical neutrality. During the simulations, the temperature was kept constant to 300 K using the Langevin thermostat, as well as the pressure to one atm (101.325 kPa) with the Langevin piston method (*Feller et al., 1995*). Non-bonded interactions were calculated using particle-mesh Ewald full electrostatics (grid spacing < 1.2 Å) (*Darden et al., 1993*). A repartition mass protocol (*Hopkins et al., 2015*) was used increasing the masses of non-water hydrogen atoms by a factor of 3 transferring mass from the heavy atom from they were bounded, allowing us to use an integration time of 4fs. A smooth 0.8–0.9 nm cutoff of van der Waals forces was employed. All MD simulations were carried out in an isobaric-isothermal (NPT) ensemble using the software NAMD 2.12 (*Phillips et al., 2005*) with the CHARMM36 force field (*MacKerell et al., 1998*; *Best et al., 2012*). Two K + ions interleaved by two water molecules were kept at the selective filter for the channel alone system; meanwhile, a single K + at position S3 escorted by two water molecules were set for the channel-toxin system, it allowing ε-amine of CTX-Lys27 occupies the S1 binding site. All systems were independently subjected to 20,000 steps of energy minimization followed by 60 ns of equilibration simulations, before beginning 400 ns of application of an external constant electric field, it perpendicular to the membrane plane to mimic a 0.5V, and 0.1V transmembrane potential (*Vergara-Jaque et al., 2019*). Two unrestrained replicas of the 0.5V were performed utilizing different initial configurations as a starting point. An additional 100 ns replica was simulated with harmonic restraints applied to the secondary structure of the toxin, as well as to the specific amino acid contacts described by *Banerjee et al. (2013)*: T8_G353: 5.7 Å, T9_G353: 5.9 Å, R25_G353: 4.8 Å, R25_D359: 8.5 Å, M29_D375: 5.7 Å, N30_D375: 5.1 Å, R34_D353: 8.7 Å, R34_D375: 8.0 Å, Y36_D357: 4.5 Å, Y36_V377: 6.5 Å. K27_Y373: 1.8 Å. All atomic systems were built and analyzed by the VMD software and the electrostatic potential maps were calculated using VMD's PMEPot plugin (*Humphrey et al., 1996*; *Aksimentiev and Schulten, 2005*).

## Acknowledgements

We thank, Victoria Prado for *Xenopus* care and oocyte preparation. Fondecyt postdoctoral grants #3170599 and #3160321 for ID-F and HM, respectively. HP was financed by Fondecyt #1171155 and the Millennium Nucleus of Ion Channel-Associated Diseases (MiNICAD), which is a Millennium Nucleus supported by the Iniciativa Científica Milenio of the Ministry of Economy, Development and Tourism (Chile). Experiment in *Figure 4* was made in 1996 in Chris Miller lab, Brandeis University.

## Additional information

### Funding

| Funder | Grant reference number | Author |
|---|---|---|
| Fondo Nacional de Desarrollo Científico y Tecnológico | 3160321 | Hans Moldenhauer |
| Fondo Nacional de Desarrollo Científico y Tecnológico | 3170599 | Ignacio Díaz-Franulic |
| Fondo Nacional de Desarrollo Científico y Tecnológico | 1171155 | Horacio Poblete |
| Ministerio de Economía, Fomento y Turismo | MiNICAD | Horacio Poblete |

The funders had no role in study design, data collection and interpretation, or the decision to submit the work for publication.

### Author contributions

Hans Moldenhauer, Conceptualization, Data curation, Formal analysis, Methodology, Writing—review and editing; Ignacio Díaz-Franulic, Conceptualization, Formal analysis, Investigation, Writing—review and editing; Horacio Poblete, Formal analysis, Supervision, Investigation, Methodology, Writing—review and editing; David Naranjo, Conceptualization, Data curation, Formal analysis, Supervision, Funding acquisition, Investigation, Methodology, Writing—original draft, Project administration, Writing—review and editing

### Author ORCIDs

David Naranjo (iD) https://orcid.org/0000-0003-3482-5126

### Decision letter and Author response

Decision letter https://doi.org/10.7554/eLife.46170.017
Author response https://doi.org/10.7554/eLife.46170.018

## Additional files

### Supplementary files

• Transparent reporting form
DOI: https://doi.org/10.7554/eLife.46170.013

### Data availability

Data used for Figures 2–7 are available at https://doi.org/10.5061/dryad.0p77qk4.

The following dataset was generated:

| Author(s) | Year | Dataset title | Dataset URL | Database and Identifier |
|---|---|---|---|---|
| Naranjo D, Moldenhauer H, Díaz-Franulic I, Poblete H | 2019 | Data from: Trans-toxin ion-sensitivity of charybdotoxin-blocked potassium-channels reveals unbinding transitional states | https://doi.org/10.5061/dryad.0p77qk4 | Dryad Digital Repository, 10.5061/dryad.0p77qk4 |

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
