## [Decision Letter]

Thank you for submitting your article "Trans-toxin ion-sensitivity of charybdotoxin-blocked potassium-channels reveals unbinding transitional states" for consideration by *eLife*. Your article has been reviewed by three peer reviewers, including Leon D Islas as the Reviewing Editor and Reviewer #1, and the evaluation has been overseen by Richard Aldrich as the Senior Editor. The following individuals involved in review of your submission have also agreed to reveal their identity: Christopher Miller (Reviewer #2) and Jon T Sack (Reviewer #3).

The reviewers have discussed the reviews with one another and the Reviewing Editor has drafted this decision to help you prepare a revised submission.

This manuscript presents calculations and experiments that provide evidence of the existence of a transient, or wobble, state during toxin unbinding from blocked Shaker channels. This is an interesting observation with potentially broad implications for other unbinding reactions. The reviewers agree that the wobble mechanism for toxin dissociation proposed in this manuscript is plausible and the authors provide evidence that it may exist. However, several comments are appended below that the authors need to seriously consider and address.

Essential revisions:

The mechanism is not well described and as currently presented may be an oversimplification. A major problem is that the Discussion gives the impression that the short-lived bound "wobble" states in Figure 8, create an ion equilibration that somehow persists in the blocked state (blocked meaning bound but not wobbled, no communication between external ions and the selectivity filter). This only seems possible while the channels remain closed. If the intracellular gate were open, wouldn't permeant ions re-equilibrate with the intracellular solution after the nanosecond wobble is complete? Figure 7E demonstrates that extracellular K^+^ has an equal or larger effect on toxin dissociation from open channels than closed channels. This suggests that the permeant ion equilibration with the external solution has little impact on ion concentrations within the selectivity filter once the wobble has ended. Please clarify whether the wobble mechanism involves memory of extracellular wobble equilibration that persists in the open blocked and/or closed blocked states.

Additionally, it is unclear how a nanosecond of external ion equilibration followed by milliseconds of internal ion equilibration with open channels could dramatically impact steady state Popen (subsection “Unresolved transitions”, second paragraph).

A more quantitative approach to the wobble mechanism is also suggested. Simple calculations of encounter complex dynamics (perhaps using equations derived in the manuscript but never deployed) could place the claims on solid footing. Such calculations could also provide estimates of how many times CTX wobbles before it fully dissociates. The equations derived assume that the blocked<->wobbled equilibrium is rapid relative to the final dissociation step, but this does not have to be the case. Instead the mechanism only requires the wobble->blocked rate to be similar to or faster than the wobble->unbound transition, to significantly impact koff.

Further quantitative discussion of voltage dependency possibly resulting solely from the wobble->blocked transition could be illuminating. The encounter complex presents a mechanism by which the phenomenon of permeant ions electrostatically pushing the toxin away could be replaced with competition between K^+^ and Lys27 for the S1 site.

The manuscript could also benefit from additional discussion of how well encounter complexes in other protein-protein systems are understood generally, and if there is any other system where such measurements have been made to this degree of precision.

In general the manuscript needs a thorough revision of grammar and language use. It is difficult to read and contains many contradictory and incorrectly constructed sentences. The exposition around Equation 2 needs to be clearer in order to understand the argument.

There is no such thing as a "binding/unbinding" equilibrium constant. You must choose whether to refer to association or dissociation. It is clear that K(K), Ko, and K∝ are association constants, since K(K) decreases as [K^+^] rises. But in the same equation, KK is a dissociation constant (as well as a notational nightmare, as it means something different and has different units than K(K)!) You must reformulate the equation so that the equilibrium constants are all of the same type.(Equation 2). The equation is correct but uses self-contradictory notation. K(K) is an explicit function of K^+^ concentration, varying between the limits Ko and K∝. But K∝ is the value when [K^+^]=0 and Ko is the value as [K^+^] approaches infinity!(Subsection “Toxin binding equilibrium is sensitive to the ion composition of the external solution”, last paragraph). The subscript-notation for the rate constants is exactly the opposite of that for the equilibrium constants. That is, koff,0 now properly refers to the value at [K^+^]=0, etc. Moreover, the word "increase" in the aforementioned subsection must be replaced by "decrease", at least in the context of the sentence, which suddenly switches to referring to the CTX dissociation constant.

KDC and KDO are claimed to be toxin 'binding constants', a word that traditionally is synonymous with association constant. But as Equation 4 makes clear, these are actually dissociation constants, not binding constants.

Given that the MD simulations are presented as evidence that the potential profiles in the selectivity filter are (very) different with and without toxin bound, the manuscript was seen by an expert in MD and several problematic statements were identified.

The calculation methods and analysis are not sufficiently explained for a general reader to understand the arguments. Details are needed on how the electrostatic potential was calculated.

It is not clear that the existence of different voltage profiles in bound or unbound channels is justified by the computational data presented.

A first problem is that the maps/profiles in Figure 2A-F appear to reflect not only the potential the ions experience due to the structure of the system, but also the potential the ions create when they're bound to the channel; at least this is what can be deduce from the peaks at z ~ 12-24 Å, along the pore axis. E.g. for the toxin-free channel (Figure 2E, black), this peak must reflect the bound ions. How many ions are included in each case and where they are isn't specified. Not knowing exactly what contributes to these maps/profiles, it is difficult to interpret them clearly. What the authors seem to do in Figure 2G is to try to isolate the effect of the applied voltage, by subtracting the potential profiles calculated at -0.5 and 0 V. In principle this might remove the contribution of the bound ions, provided that the occupancy of the pore is the same in both cases – but this is not a given since the application of voltage is said to induce movement of ions in the simulations. So it is not clear what to make of the data in Figure 2G either. But taking this data at face value, what the plot shows is that an ion at z = 12 moving to z = 24 (which appears to be the selectivity filter – see Figure 2E, black) would experience a significant driving force outwards as a result of the applied potential, when the toxin is bound to the channel (red curve); the same effect is seen for the unblocked channel, logically (black curve). So this result seems to contradict the authors claim.

In summary, the reviewers agree that the computational section requires a re-evaluation, which might or might not be favorable. First, it is crucial that methodology is presented and rationalized more clearly, to enable the reviewers to assess whether it is appropriate for the question at hand. In addition, the results ought to lead to a clear-cut interpretation that supports the conclusions put forward.

---

## [Author Response]

Essential revisions:The mechanism is not well described and as currently presented may be an oversimplification. A major problem is that the Discussion gives the impression that the short-lived bound "wobble" states in Figure 8, create an ion equilibration that somehow persists in the blocked state (blocked meaning bound but not wobbled, no communication between external ions and the selectivity filter). This only seems possible while the channels remain closed. If the intracellular gate were open, wouldn't permeant ions re-equilibrate with the intracellular solution after the nanosecond wobble is complete? Figure 7E demonstrates that extracellular K^+^ has an equal or larger effect on toxin dissociation from open channels than closed channels. This suggests that the permeant ion equilibration with the external solution has little impact on ion concentrations within the selectivity filter once the wobble has ended. Please clarify whether the wobble mechanism involves memory of extracellular wobble equilibration that persists in the open blocked and/or closed blocked states.

The reviewers are right. We totally overlooked this aspect. To explain the lack of state dependency in the trans-toxin ion sensitivity, in this version, in the sixth paragraph of the subsection “Unbinding dynamics”, we discuss the possibility that only the most external aspects of the pore normally reach equilibrium. Thus, the impact of the internal activation gate is minimized.

Additionally, it is unclear how a nanosecond of external ion equilibration followed by milliseconds of internal ion equilibration with open channels could dramatically impact steady state Popen (subsection “Unresolved transitions”, second paragraph).

In the second paragraph of the subsection “Unresolved transitions”, we discuss that if such a large shift in the apparent open probability really existed, we should have been able to detect ionic currents near -70 mV or less. Nevertheless, because we did not detect such currents, we favor the idea that the negatively shifted voltage dependency of the apparent Po represents the probability that the electric field along the pore has been reestablished during wobbling.

A more quantitative approach to the wobble mechanism is also suggested. Simple calculations of encounter complex dynamics (perhaps using equations derived in the manuscript but never deployed) could place the claims on solid footing. Such calculations could also provide estimates of how many times CTX wobbles before it fully dissociates. The equations derived assume that the blocked<->wobbled equilibrium is rapid relative to the final dissociation step, but this does not have to be the case. Instead the mechanism only requires the wobble->blocked rate to be similar to or faster than the wobble->unbound transition, to significantly impact koff.

In section “unresolved transitions” we discuss the time scale of the local unbinding events based. The reviewers correctly consider that theoretically time scale of wobbling does not need to be much faster than that of unbinding. However, if that were the case, given that toxin unbinding takes from milliseconds to hundreds of seconds and, given that wobbling restores the electric field along the pore, those events would be electrophysiologically detectable in single channel recordings.

Further quantitative discussion of voltage dependency possibly resulting solely from the wobble->blocked transition could be illuminating. The encounter complex presents a mechanism by which the phenomenon of permeant ions electrostatically pushing the toxin away could be replaced with competition between K^+^ and Lys27 for the S1 site.

We thank the reviewers for this comment. In the discussion of Equation 8 we added a reference to the competitive nature of the Lys27 rebinding to S1. We reasoned that if both equilibria, described in Equations 2A and 2B, correspond to binding (kon) and competitive rebinding of Lys27 (affecting koff), then the ratio of their respective dissociation constants put lower limit to the probability that S1 is empty during wobbling.

The manuscript could also benefit from additional discussion of how well encounter complexes in other protein-protein systems are understood generally, and if there is any other system where such measurements have been made to this degree of precision.

In the last paragraph we discuss this matter by comparing protein unbinding with protein unfolding. There are recent and old examples of rare and ephemeral intermediate states during protein folding (Raschke and Marqusee, 1997; Tapia-Rojo et al., 2019) or during protein oligomerization (Maity et al., 2018). Nevertheless, it remains to be shown if states that arise while proteins are mechanically stretched can be spontaneously visited.

In general the manuscript needs a thorough revision of grammar and language use. It is difficult to read and contains many contradictory and incorrectly constructed sentences.

We have extensively revised the grammar and language along the manuscript. We believe that this time it is much better written.

The exposition around Equation 2 needs to be clearer in order to understand the argument.

We rewrote that part and divided Equation 2 in two parts to describe separately how k_on_ is antagonized and K_off_ is enhanced by external K^+^. This action reduced the confusing notation

There is no such thing as a "binding/unbinding" equilibrium constant. You must choose whether to refer to association or dissociation. It is clear that K(K), Ko, and K∝ are association constants, since K(K) decreases as [K^+^] rises. But in the same equation, KK is a dissociation constant (as well as a notational nightmare, as it means something different and has different units than K(K)!) You must reformulate the equation so that the equilibrium constants are all of the same type.Done. See previous answer.(Equation 2). The equation is correct but uses self-contradictory notation. K(K) is an explicit function of K^+^ concentration, varying between the limits Ko and K∝. But K∝ is the value when [K^+^]=0 and Ko is the value as [K^+^] approaches infinity!Done. See previous answer.(Subsection “Toxin binding equilibrium is sensitive to the ion composition of the external solution”, last paragraph). The subscript-notation for the rate constants is exactly the opposite of that for the equilibrium constants. That is, koff,0 now properly refers to the value at [K^+^]=0, etc. Moreover, the word "increase" in the aforementioned subsection must be replaced by "decrease", at least in the context of the sentence, which suddenly switches to referring to the CTX dissociation constant.Corrected.KDC and KDO are claimed to be toxin 'binding constants', a word that traditionally is synonymous with association constant. But as Equation 4 makes clear, these are actually dissociation constants, not binding constants.

Corrected: KDC and KDO are dissociation constants along the manuscript.

Given that the MD simulations are presented as evidence that the potential profiles in the selectivity filter are (very) different with and without toxin bound, the manuscript was seen by an expert in MD and several problematic statements were identified.

We thank the reviewers for highlighting the importance of the MD simulations, as well as for searching for an expert on the field. We believe that in this aspect is where the manuscript shows most of the improvements.

The calculation methods and analysis are not sufficiently explained for a general reader to understand the arguments. Details are needed on how the electrostatic potential was calculated.

We agree with the comment, and we went deeper on the explanation of the MD details. We improved the description of the methodologies used on the MD section, specifically on the section “Molecular Dynamics simulations show wobbling”, as well as in the section Materials and methods “Molecular Dynamics (MD) Simulations”.

It is not clear that the existence of different voltage profiles in bound or unbound channels is justified by the computational data presented.

We agree with the reviewers’ comment. We realized that the scope of the manuscript was not to interpret the voltage profiles along with the MD simulations. Instead, we explored the wobbling effect by following the position of CTX under different voltages, specifically by measuring the average distances separating pair contacts between at the bottom of the interaction surface.

Additionally, we ran several replicas under internally applied potential, as well as extended the simulations until 400ns. It was possible to identify the voltage profiles in the absence of CTX, when the toxin was bound, and when it was wobbling.

On the other hand, we included an extensive section describing atomistic details of the movement of CTX (wobbling) on the simulations under external voltage.

A first problem is that the maps/profiles in Figure 2A-F appear to reflect not only the potential the ions experience due to the structure of the system, but also the potential the ions create when they're bound to the channel; at least this is what can be deduce from the peaks at z ~ 12-24 Å, along the pore axis. E.g. for the toxin-free channel (Figure 2E, black), this peak must reflect the bound ions. How many ions are included in each case and where they are isn't specified. Not knowing exactly what contributes to these maps/profiles, it is difficult to interpret them clearly. What the authors seem to do in Figure 2G is to try to isolate the effect of the applied voltage, by subtracting the potential profiles calculated at -0.5 and 0 V. In principle this might remove the contribution of the bound ions, provided that the occupancy of the pore is the same in both cases – but this is not a given since the application of voltage is said to induce movement of ions in the simulations. So it is not clear what to make of the data in Figure 2G either. But taking this data at face value, what the plot shows is that an ion at z = 12 moving to z = 24 (which appears to be the selectivity filter – see Figure 2E, black) would experience a significant driving force outwards as a result of the applied potential, when the toxin is bound to the channel (red curve); the same effect is seen for the unblocked channel, logically (black curve). So this result seems to contradict the authors claim.

We thank the reviewers for bringing these issues on the voltage profiles section; for reference, Figure 2 has been removed from the revised paper. We abandoned the idea that a detail description of the electrostatic profiles could provide insights of electric field across the pore. Regarding the electrostatic calculations, in this version, we pursue the idea that the electrostatic potential profile during Lys27 wobbling looks more similar to that of the toxinfree channel than to the tight blocked one. Now, we include a detailed description of the initial configuration of the simulated systems. Additionally, as we decided to improve the atomistic description of CTX wobbling, we moved the description of the voltage profiles to the supplementary information.